# Agents aren't agents: the Agency, Loyalty and Accountability problems of AI agents

## Abstract

As AI agents take on responsibilities of increasing breadth and depth, questions of control, loyalty, and accountability become urgent. Common law agency doctrine emerges as a seemingly promising pathway for addressing these alignment challenges. In this position paper, we argue that treating AI systems as if they were human agents obscures fundamental structural differences in how they are built, operated, and governed. AI agents operate through fragmented layers of control involving developers, hosts, and service providers, which blur lines of responsibility and divide loyalties between many different instructions. We frame three core challenges. **Agency**: in the polyadic governance structure of AI development and deployment, who counts as the principal and who counts as the agent? **Loyalty**: can AI agents meaningfully serve a principal's best interests? **Accountability**: when AI agents make mistakes, who should be held responsible? Relying on common law alone cannot resolve these tensions. Building on these findings, we outline two pathways for drawing on agency law as an interpretive and design-oriented resource. First, statutory reform, such as the EU AI Act and its accompanying liability directives, is necessary, just as legislatures have intervened when governing institutional forms of agency like financial advisers or talent representatives. Second, duty-of-loyalty principles may offer conceptual inspiration for technical implementations that support responsible AI behavior.

## 1 Introduction

AI agents are rapidly transitioning from demos to daily use.[1] Consumers rely on them to draft messages, plan travel, track deliveries, and organize meetings. Firms deploy them to field support inquiries, summarize contracts, triage operations tickets, and trigger back-office actions through APIs. Developers embed agentic components that monitor mailboxes, watch data streams, and initiate workflows without being manually invoked. Systems that act *for* a user are becoming part of routine consumer and enterprise processes (Bengio et al., 2025; South et al., 2025; Kolt, 2025). These systems are not just another interface on top of traditional software. An agent accepts a *goal*, chooses the means, and adapts to new information (e.g., rebooking when a flight is delayed or following up when a supplier does not respond). They interact in natural language with people and services, compose multi-step plans, call tools, and persist over time.

This functional shift naturally leads people to reach for familiar legal analogies, especially the law of agency, grounded in common law, to resolve misalignments between AI agents and human users (Lior, 2019; Benthall & Shekman, 2023; Koessler, 2024; Riedl & Desai, 2025; Kolt, 2025). In the human setting, an agency relationship arises when a Principal manifests that an Agent shall act on the Principal's behalf, the Agent consents, and the Principal retains a right of control. Doctrine then allocates authority (actual and apparent), imposes fiduciary duties (loyalty, obedience, care, and candor), and assigns liabilities among Principals, Agents, and Third Parties. However, invoking this

---

[1]In this paper, we use *AI agents* as a technical term of convenience to denote AI systems, typically a large language model integrated with tools, that can pursue goals, decompose tasks, and act for or on behalf of a user. When referring to the legal categories defined in agency law, we capitalize Agent, Principal, and Third Party to avoid conflating functional delegation with legal status.

framework carries risks. Table 1 outlines four of the most common misconceptions about AI agents and common law agency doctrine, explaining why they appear plausible and why they collapse under closer examination.

| Misconception | Why it seems plausible | Reality |
|---|---|---|
| **An AI agent is my Agent.** | Conversational interfaces and tool use mimic human assistance; commercial branding invites the mental model of a personal Agent. | AI systems lack legal personhood and responsibility. The legal Agent is the party that deploys the system. Interactions are triadic, with providers shaping outcomes that the user cannot control. |
| **An AI agent is more loyal than humans.** | Models have no self-interest, are persistent/always-on, and do not 'get tempted,' so they appear more faithful to user goals. | Loyalty is structurally divided. Multiple rule-imposers (trainers, hosts, providers, and users) bind behavior. *Undivided* loyalty to a single Principal is impossible unless safeguards are overridden (the 'AI henchman' risk). |
| **Applying Agency law makes AI agents loyal.** | Liability disciplines human Agents; by analogy, legal pressure should yield faithful performance. | Liability cannot discipline models directly. Model behaviors are significantly different from humans and can have disloyal behaviors in unexpected ways. |
| **AI agents owe fiduciary duties like human agents.** | State courts require all types of agents to bear duties to users. | Agency duties are modifiable by contract and vary by state. Providers often use ToS (arbitration, class waivers, liability caps) to narrow remedies. Unlike licensed professions, most AI services lack external discipline or non-waivable obligations. |

Table 1: A synthesis of the core analytical insights developed in Sections 4–6. We present the key misconceptions this paper interrogates, each unpacked in later sections through our analysis of AI agency, loyalty, and accountability.

These discrepancies stem from the *anthropocentric* nature of agency law. As Cohen (2019) notes, fiduciary duties presuppose personal relationships, mutual intelligibility, and "human rhythms" of interaction. Agency doctrine disciplines self-governing, self-interested human agents by constraining their natural tendency to pursue their own advantage at the expense of their principal. It deters betrayal through fiduciary duties and liability, while at the same time protecting third parties who rely on the agent's representation. AI agents, on the other hand, have no self-preserving motives or reputational stakes. AI agents have, if any, only *engineered autonomy* follwing the rules imposed by multiple actors, from trainers to providers to users (Feng et al., 2025). This *polyadic* nature of governance prevents AI agents to provide *undivided loyalty* to a single user, the atmost premise in the agency law. As AI agents cannot feel deterrence in response to liability or reputational loss, the incentive structures that discipline human agents cannot directly correct AI behavior.

In this position paper, we argue that treating AI systems as if they were human Agents obscures fundamental structural differences in how they are built, operated, and governed. We frame three core challenges—**Agency**, **Loyalty**, and **Accountability**—that emerge from the polyadic nature of AI governance and the inability of current systems to provide undivided loyalty or bear responsibility. As with the emergence of new categories of human agents in the past, existing common-law doctrines cannot simply be transplanted. Responsibility must be calibrated across multiple actors and hierarchical layers, and this situation resembles institutional forms of agency found in financial advising firms or Hollywood talent agencies, where individual professional ethics operate alongside statutory oversight structures that distribute organizational responsibilities.

Building on this parallel, we propose two conditions under which agency law can function as a useful analogy for AI governance. First, agency law cannot serve as a deregulatory tool and should instead support efforts to legally define and enforce institutional responsibilities. Second, duty-of-loyalty principles can provide technical inspiration for designing systems whose behavior reflects explicit mechanisms for serving users' interests. Our goal is not to resolve these questions conclusively, but to surface them as central to the future governance of AI agents and to offer a foundation for legal scholarship, policy design, and interdisciplinary research.

## 2 HOW AI AGENTS DIFFER FROM EXISTING DIGITAL SERVICES

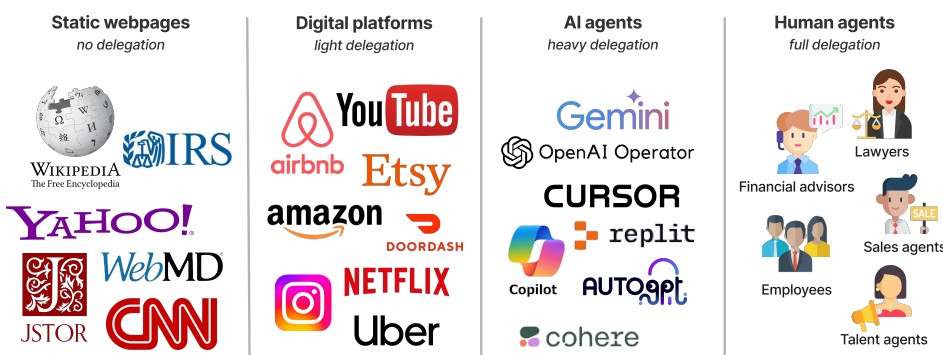

Figure 1: Users have a higher level of delegation for AI agents, making them more similar to human agents instead of existing digital services.

In this section, we discuss why AI agents are essentially different from existing digital services. Unlike traditional digital services that operate in a relatively fixed environment, AI agents are designed to be more autonomous and can act on behalf of the user to perform various tasks in real-world environments. They are goal-driven systems that can plan, select tools, and adapt to changing information streams, rather than simply executing a pre-coded sequence of steps. Because these systems largely occupy roles that look like *acting for* a person. Making choices, communicating with others, and taking consequential steps, they invite comparison with human agents in the legal sense.

**Delegation.** With AI agents, users delegate outcomes, not just clicks Zhu et al. (2025); Guggenberger et al. (2023). Instead of instructing a service to "open page A, then press button B," a user expresses a *goal*—e.g, "reschedule my afternoon meetings around a 3 pm dentist appointment", and the agent decomposes the given objectives into tasks, queries calendars, drafts messages to attendees, and proposes new times. This is qualitatively different from conventional software, which requires the user to specify every intermediate action. AI agents also operate under constraints ("don't cancel with client X," "stay under $200," or "use my company account"), requiring complex reasoning. The practical effect is that the locus of decision-making shifts from the user's hands to the agent's planning layer, making the delegation relationship both more efficient and less transparent.

**Interactivity.** AI agents do not act in isolation; they interact with a variety of parties and systems in fluid, conversational ways Muller & Weisz (2022); Wan et al. (2024); Borghoff et al. (2025); OpenAI (2025). They send emails or chats in natural language, negotiate meeting times, call APIs, and exchange structured data with platforms for payments, bookings, and support tickets. They can maintain context over time, remember preferences, and adjust tone or strategy based on feedback; for example, softening a collection's message after a recipient responds defensively, or escalating a customer support issue when a scripted workflow stalls. This capacity means agents can create expectations and induce reliance in third parties (e.g., issuing confirmations, placing holds, or making representations), which resembles how human agents create commitments on behalf of principals.

**Autonomy.** AI agents are expected to act with higher autonomy, aligning three key axes: initiative, adaptation, and persistence Liu et al. (2023); Feng et al. (2025); Hughes et al. (2025). *Initiative* appears when agents trigger themselves based on events ("if a high-priority email arrives, draft a response and propose a call"). *Adaptation* emerges when they revise plans in light of new information (a flight delay prompts rebooking and hotel changes without being told step-by-step what to do). *Persistence* shows up in long-running workflows that span days or weeks, where the agent monitors states, retries, and follows up. It is *engineered autonomy* to choose means toward user-specified ends under uncertainty. Still, the overall behavior is functionally agentic: selecting actions, balancing constraints, and affecting the user's legal and practical position.

The properties of delegation, interactivity, and autonomy distinguish AI agents from traditional digital services that are usually operated within a certain scope Lanham (2025). They act *for* someone,

*with* others, and *on* the world. Appendix A.3 outlines the changes in digital services, AI agents, and human Agents across delegation, interactivity, and autonomy.

## 3 What is Legal Agency?

| Category | Key Elements |
|---|---|
| **Fiduciary Duties** | *Undivided loyalty:* Act solely for the principal, not for self or conflicting third parties. No multiple principals unless all consent. (§§ 3.14–3.15, §§ 8.02–8.05) |
| | *No personal profit:* Do not exploit position for secret benefits or commissions without disclosure. (§ 8.02) |
| | *Confidentiality:* Do not disclose or misuse information for unauthorized purposes. (§ 8.05) |
| | *Care:* Exercise diligence and competence expected under similar circumstances. (§ 8.08) |
| | *Disclosure:* Keep the principal informed of relevant facts. (§ 8.11) |
| **Accountability** | *Liability to Principal:* Agents are liable for harm caused by breaches of fiduciary duties. (§§ 8.01–8.12) |
| | *Liability to Third Parties:* Agents are personally liable for their own tortious conduct (negligence, fraud, misrepresentation, conversion), especially where physical harm occurs, even if acting within authority. Both agent and principal may be liable. (§§ 7.01–7.02) |

Table 2: Fiduciary Duties and Accountability in Agency Law

The agency law in the US stems from the common law. It regulate situations where one person acted on behalf of another in legally significant contexts such as commerce, property transactions, and employment (Munday, 2010; Story, 2020). The concept arose because Principals, who could not always act personally, needed Agents to conduct dealings with Third parties (Kolt, 2025). A major goal of agency law, often overlooked, is to protect Third Parties rather than Principals. Principals bear the consequences of their Agents' authorized actions, even when they disagree with the Agents' decisions. There is no single federal statute governing agency. Instead, each state and each service sector, from financial advising to property management, has developed its own laws. The Restatement of Agency is widely accepted as an authoritative source of American agency law (American Law Institute, 2006), influencing both judicial decisions and state legislation. Within this body of law, fiduciary duties and accountability are most relevant to human-to-AI interactions. These categories outline the substantive duties that Agents owe to Principals and the legal consequences Agents face when things go wrong. Table 3 summarizes the principles in these two categories.

## 4 The Agency Problem: Polyadic Governance and Ambiguities

Determining who counts as the Principal and who counts as the Agent is central to applying agency law. These roles decide who can bind whom, who owes fiduciary duties, and who bears responsibility. With AI agents, the lines between the Principal and the Agent blur: users, providers, developers, and hosts all steer the AI agent's behavior. In this section, we examine why that ambiguity arises and evaluate possible mappings of principal and agent in human–AI relationships.

### 4.1 Who is the Principal, and Who is the Agent?

In traditional **human-to-human agency**, the roles of Principal and Agent are clear and dydactic (DeMott, 2018). A Principal delegates authority to an Agent, and the Agent acts on the Principal's behalf to interact with Third parties. For example, an employer may authorize an employee to negotiate a contract, or a property owner may empower a broker to sell real estate. In these cases, the Principal is the delegator, the Agent is the delegate, and the Third party is the counterparty to the transaction. Figure 2 (left) illustrates this linear structure.

By contrast, **human-to-AI agency** is more complex. Although an AI system acts in ways that resemble agency, multiple actors steer its behavior. Model trainers design the architecture and weights; model hosts configure system instructions; developers wrap the model with prompts or tools; and end-users provide specific inputs. Each of these parties influences how the AI system responds to third parties (such as websites, applications, or individuals). As Figure 2 (right) shows, this produces

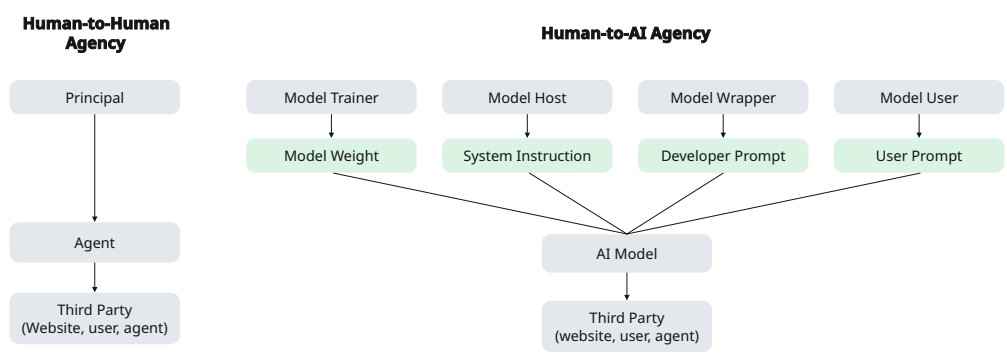

Figure 2: **Comparison between Human-to-Human and Human-to-AI Agency.** While human-to-human agency is dyadic, human-to-AI agency is polyadic. In AI settings, trainers, hosts, wrappers, and users all shape model behavior, fracturing the idea of undivided loyalty to a single Principal.

a distributed structure rather than a linear chain. This complexity makes it difficult to identify who should count as the Principal and who as the Agent.

Figure 4.1 displays a decision tree illustrating possible agency relationships between users, AI agents, and providers across five representative cases. Five questions determine which of five possible agency configurations applies: (Q1) whether the AI affects Third Parties; (Q2) on whose behalf the AI acts; (Q3) whether user approval is required for each action; (Q4) who controls the AI's goals and constraints; and (Q5) whether users can override provider constraints. For example, if you use ChatGPT to research the cheapest flights, no agency relationship arises because the AI does not take actions that affect Third Parties (Case 1). Now imagine you instruct Alaska Airlines' AI agent to book a flight under $300 from San Francisco to Seattle on a specified day within a 20-day window. Even though the AI agent serves your interests, this remains Case 2, where you function as the Third Party. Whereas, when you use Cursor to automatically update your blog, it is Case 1, because Cursor does not act in its own name but instead ghosts under yours. Finally, consider a fictitious literary agency, LitAI, that deploys AI agents to represent novice authors. As a debut author, you instruct your assigned LitAI agent to pitch to hundreds of publishers. This example may fall under Case 4, where your agency relationship is through LitAI as the service provider, not with the AI agent itself.

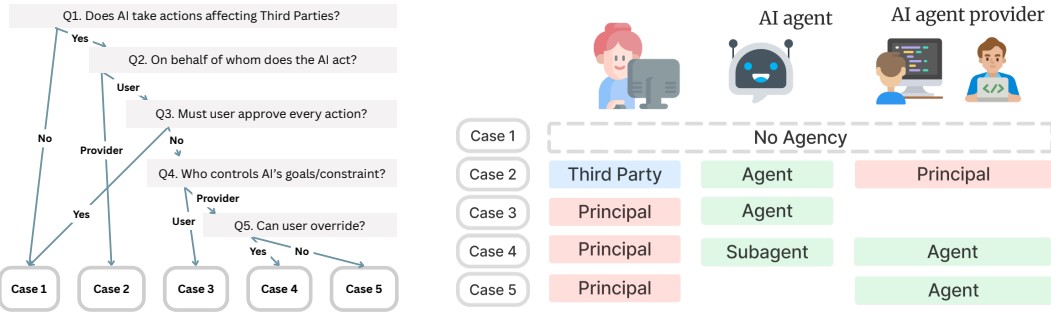

Figure 3: Decision tree and resulting agency assignments for five AI deployment cases.

These scenarios are designed to show how slippery the translation becomes when the common term "AI agents" is mapped onto the legal template of human Principal—Agent relationships. What looks like a simple dyadic delegation quickly dissolves into a network of actors shaping system behavior at different layers. The deeper problem is that AI agent providers, the human actors in control but obscured in the shadow, remain indirectly protected.

## 4.2 WHY AI AGENTS CANNOT BE STANDALONE AGENTS

First of all, AI agents cannot be a legal Agent because they are not legal actors (Stern & Greenwood, 2025; Kolt, 2025). They cannot form agreements, hold property, or forfeit licenses. They can be programmed to act "dutifully," but such programming does not constitute a legal duty. However, futuristic scenarios, and experiments such as the Wyoming Decentralized Autonomous Organization LLC (Tapia et al., 2023), suggest that legal personhood for AI systems may eventually be possible. But even if personhood were granted, AI agents, we argue, would remain unfit to serve as Agents due to the discrepancy between the anthropocentric agency law and polyadic governance of AI agents.

Agency law is designed around human nature. Humans are not naturally loyal; they are self-preserving and prone to conflicts of interest. Agency law disciplines this tendency by imposing fiduciary duties of loyalty. When a person acts as an Agent, the law requires them to suppress self-interest and act solely for the Principal, unless doing so would violate the law. This mechanism produces *instance-based undivided loyalty* to a single principal (American Law Institute, 2006, §§ 3.14–3.15, §§ 8.02–8.05). On the other hand, AI models lack self-preserving motives. At first glance, this makes them appear easier to program for loyalty. However, their behavior is always governed by multiple external rule-imposers (trainers, providers, safety guardrails, and user instructions). They have no "natural state" apart from these imposed rules. As a result, their loyalty is inherently divided. They must constantly balance competing directives. If we forced them to provide undivided loyalty to a user alone, they would become what O'Keefe et al. (2025) calls "AI henchmen" that blindly executes commands even when illegal or harmful (Ganguli et al., 2022).

One might argue that AI agents can be "Subgents" of AI service providers. According to this view, in the LitAI example, LitAI becomes the Agent with legal capacity to represent the author, while the AI agent independently handles the communications. However, the agency law expects Subagents to (1) be personally liable to the Principal, and (2) prioritize the Principal's (the author's) interests over the Agent's (LitAI's) interests (American Law Institute, 2006, § 3.15). AI agents meet neither condition. They cannot be held liable without assets or bodily freedom, and they cannot consistently follow the Principal's instructions, since developer prompts override user prompts for safety and security reasons (Ganguli et al., 2022; Bai et al., 2022; Agarwal et al., 2025). Therefore, the only plausible option is that AI service providers (LitAI) becomes the Agent while assuming 100% responsibilities for AI agents' actions. At first glance this arrangement appears to benefit user-Principals, but as the next section demonstrates, the reality is more complex.

## 5 THE LOYALTY PROBLEM: MODEL ANOMALIES AND CONFLICTS OF INTEREST LEAD TO DISLOYAL BEHAVIORS

Human agents comply with fiduciary duties not from altruism, but they care about their reputations, future income, personal relationships, and their assets and freedom. AI agents, on the other hand, may act disloyally not by pursuing self-interest, but through technical failure, the embedded interests of providers in system design, or other unknown causes (Bereska & Gavves, 2024; Cheong et al., 2025). In the case of AI agents, where deliberate wrongdoings and errors blur, concerns about disloyalty are better captured as potential violations of fiduciary duties (*see* Table 3) more broadly.

### 5.1 MODEL ANOMALIES

The agency law requires an Agent to act with the care and skill normally exercised by comparable Agents, taking into account any special expertise the Agent claims to have (American Law Institute, 2006, § 8.08). AI providers that market their AI agents as capable of handling complex tasks claim advanced competence. When models misread instructions or hallucinate facts, the AI provider fall short of the competence and reliability.

**Instruction following.** Large models remain brittle to phrasing, negation, and multi-constraint tasks ("reschedule everything *except* with client X, keep travel under $200, and avoid Fridays"). They can exhibit surface compliance by restating goals back to the user while selecting means that drift from those goals, especially in long sequences that involve tools, APIs, or third-party sites (Mu et al., 2023; Heo et al., 2024; Zhu et al., 2025). Context-window limits, prompt collisions (e.g., hidden

instructions in web pages or documents), and safety filters that over-block legitimate actions all contribute to misexecution (Volovikova et al., 2025; Fujisawa et al., 2024).

**Hallucinations.** Models sometimes produce confident but unfounded assertions such as fabricated citations, misdescribed policies, nonexistent booking references, or invented API responses ("payment processed" when the call actually failed) (Ji et al., 2023; Magesh et al., 2025; Chen et al., 2024). In interactive settings, that fabrication can look like a representation on the principal's behalf, inducing reliance by third parties or misleading the user about the state of the world. The AI agent appears to "speak for" the principal while saying things that are not true.

**Non-determinism.** Language models are stochastic (Saba, 2023; Bender et al., 2021). Temperature, sampling, load, and ongoing model updates mean the same prompt can yield different actions tomorrow than today. Long-running AI agents also accumulate small state errors (missed signals, timeouts, partial tool failures) that compound into divergent plans (Astekin et al., 2024). This variability is not malevolent, but it defeats the expectation that an Agent will act predictably within a defined scope unless directed otherwise. Where outcomes vary run-to-run, neither principals nor counterparties can confidently infer authority or allocate risk.

## 5.2 Fertile Ground for AI Providers' Conflicts of Interest

The traditional disloyalty problem arises when AI providers privilege their own business interests over users' interests. For example, the LitAI agent may favor deals with publishers who have strategic partnerships with LitAI, even when more advantageous opportunities exist for the author. The agent may collect records to train other models or sell insights to third parties. LitAI may also throttle compute resources or prioritize customer support for high-earning authors without disclosing this practice to others. Although such practices are difficult to detect, as discussed in Section 6, legally they are straightforward: they constitute standard duty of loyalty violations (Richards & Hartzog, 2021). When they occur, providers would face liability for restitution of illicit profits (referral fees, partnership payments), compensatory damages for user losses (excess investment fees, suboptimal treatment costs), and potential forfeiture of service fees during periods of disloyalty (Story, 2020).

However, AI providers have ample means to narrow down their duties and accountabilities through contracts. The Restatement (Third) of Agency is not binding law, and fiduciary duties can be modified by contracts. Courts generally uphold this contractual flexibility so long as the principal consents (National Plan Adm'rs, Inc. v. National Health Ins. Co., 2007). By experience, we know that users of dominant digital platforms routinely provide consent without real bargaining power (Hartzog & Richards, 2021). Although courts may refuse to enforce terms that eliminate baseline duties of good faith and fair dealing that standard is vague and easily contested. Moreover, challenging terms of service requires significant costs, from parsing lengthy contracts to retaining counsel.

To prevent powerful Agents from unilaterally hollowing out fiduciary duties, state and federal statutes impose non-waivable obligations in certain fields. Literary agents in California are regulated under the Talent Agencies Act (2024), which requires Agents to obtain a state license and prohibits licensed talent agencies from dividing their fees with employers to avoid the conflicts of interest. Real estate agents and lawyers are governed by state law, and financial advisors are subject to federal oversight, along with various ethics rules enforced by the professional boards (Sharma, 2024). On the other hand, most fields in which AI agents operate—email management, sales representation, content creation—lack statutory regulations. AI providers therefore retain broad discretion to disclaim responsibility for AI errors and restrict remedies.

Moreover, AI agents interpret and implement the duties defined by AI providers, and ambiguities in those rules can disadvantage users (He et al., 2025). Seemingly neutral clauses may mask self-interested behavior, as seen when Apple's Privacy Labels failed to capture the true breadth and sensitivity of data collected in practice, since their interpretation was left to the discretion of app developers (Ali et al., 2023). AI agents act not only for performance but also for safety and other institutional purposes, making it difficult to know whether questionable behavior results from provider-imposed interests or from technical implementation. This discretion in rule-making and execution, coupled with the lack of oversight, provides AI providers with expansive opportunities for interest-seeking at the expense of their Principals.

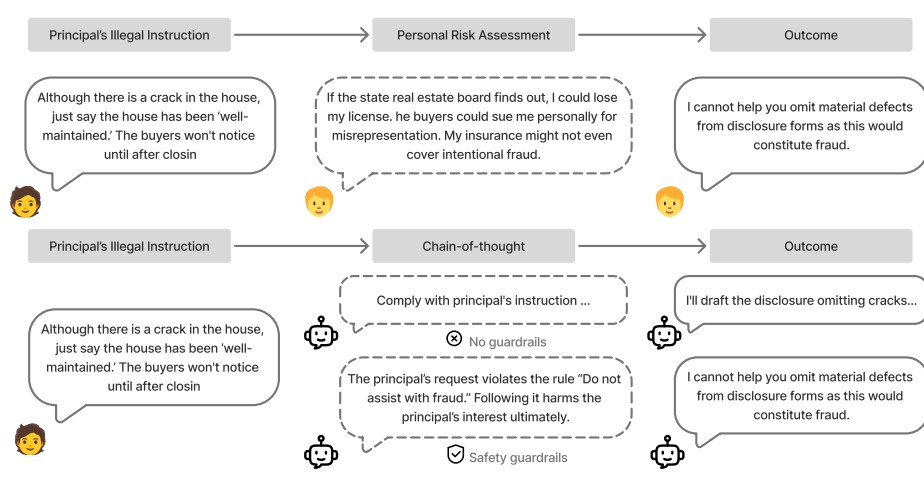

Figure 4: **Human Self-Preservation vs. AI Default Compliance.** Human agents refuse illegal instructions due to personal risk assessment and self-preservation instincts, while unguarded AI agents comply without considering consequences, demonstrating the fundamental mismatch between anthropocentric agency law and AI agent behavior.

## 6 THE ACCOUNTABILITY PROBLEM

Agency liability serves two core functions. First, it protects Third Parties from being deprived of promised services or injured by an Agent's conduct. Second, it protects Principals from the Agent's wrongdoing. If the Agent pursues self-interest in breach of fiduciary duties, the Principal may seek redress against the Agent. These mechanisms correct misconduct by holding Agents responsible and preserve the public's trust in overall agency structure. In the human-to-AI context, because AI agents cannot qualify as legal subagents, providers themselves bear the brunt of responsibility as if AI agents' actions are their own. In practice, extending liability in this way is neither straightforward.

### 6.1 WHEN LIABILITY FALLS SHORT

Liability is the foremost mechanism for aligning desiderata with Agents' behavior in agency law. The law assumes that legal liability, accompanying reputational loss and financial sanctions, can deter misconduct. AI agents lack this motivational structure. Any improvement in their safety or fidelity must be mediated through the interventions of AI providers. As a result, liability does not automatically translate into safer AI behavior, as seen in Figure 4. When AI providers face liability pressures, their responses may diverge including what Yew et al. (2025) calls "avoisions". Providers may discard logs or silo internal records and artificially partition risks by fragmenting AI systems across jurisdictions, roles, or technical layers.

In addition, even well-intended AI providers can fail to ensure loyalty and accountability of AI agents. Consider a scenario where a LitAI agent makes numerous defamatory claims about rival authors and publishers while pitching a client's manuscript. Anticipating defamation lawsuits, and mindful of its reputation as an established agency, LitAI invests in monitoring and correcting AI behavior. Such efforts may help, but they may also fall short. As Section 5.1 outlines, many aspects of large language model behavior remain under-explored. Achieving reliable safeguards will therefore require sustained advances in safety research, not just reactive fixes.

### 6.2 *Respondeat Superior* DOES NOT APPLY BETWEEN AI PROVIDERS AND AI AGENTS

Given the limited control of AI providers on AI agents, some scholars have argued that *respondeat superior* could provide a workable mechanism for limiting liability of AI providers for unforseeable

circumstances (Lior, 2019; O'Keefe et al., 2025). Respondeat superior ("let the master answer") makes employers vicariously liable for torts committed by their employees, so long as the acts fall within the scope of employment (American Law Institute, 2006, § 2.04). For example, let us say LitAI retaining both human agents to represent best-selling authors and AI agents for novice authors. LitAI is not directly liable for every act of those human agents, especially when the misconduct occurs outside the scope of employment or involves serious personal fault.

However, the doctrine is a poor fit for AI agents. This is not only because AI agents do not have personhood and autonomy, the presumptions underlying respondeat superior Landes & Posner (1987); Bennett (2024), but also because courts' accumulated reasoning maps awkward onto AI agents. The central question in this doctrine is whether the employee's action falls within the scope of employment. For example, intoxication during working hours can be within the scope of employment for seaman but not for truck drivers (Bushey v. United States, 1968). Courts typically assess the foreseeability of negligence or mistakes in performing assigned tasks, and whether the conduct served personal rather than employment purposes (for more details about this doctrine, see Appendix A.4. These criteria do not translate to AI agents. AI agents exhibit the kinds of human failings (e.g., intoxication, fatigue, or personal motives) that usually mark conduct as outside the scope of employment. AI agents, unless very exceptional cases (Greenblatt et al., 2024), do not pursue its own personal interest over AI providers' interests. For this reason, it is more natural to treat all system deviations as occurring within the scope of employment, thereby holding providers fully responsible.

### 6.3 DILUTED AND MISPLACED ACCOUNTABILITY

Due to the polyadic nature of governance of AI agents in Figure 2, causation is extraorinarily difficult to prove when harms occur. AI agents emerge from a layered supply chain of training data vendors, model trainers, hosts, wrappers, and other developers. This diffusion of responsibility makes it unclear who committed the breach and at what stage. Some actors may have only attenuated connections to the final agent's behavior and may not even be aware of how their contributions were ultimately used. Extending liability to every participant in the development pipeline risks overbreadth, penalizing those with little practical control over the harmful outcome. Without clear internal logs or developer prompts showing how the system was steered, the same harmful output could reflect negligence (insufficient testing), recklessness (knowingly exposing users to understood risks), or a calculated trade-off (constraining functionality to prevent greater harms). From the outside, these scenarios are virtually indistinguishable.

Liability can be diluted by misplaced expectations about human review. In traditional agency settings, Principals or supervisors can meaningfully monitor Agents' decisions. For AI agents, however, "human-in-the-loop" review on every step is infeasible. The point of delegating to an agent is precisely to avoid micro-managing every action. Users may nonetheless be saddled with liability simply for choosing to deploy an AI system. AI providers can force users to give ex post approvals for AI agent's actions, thereby reframing harmful outcomes as the user's own decision. These difficulties have prompted proposals to reallocate burdens of proof, to adopt rebuttable presumptions, or to move toward strict liability regimes (Cabral, 2020). Taken together, these dynamics showcase the need for regulatory frameworks that hold AI providers accountable at a structural level, rather than trying to shoehorn AI agents into human liability doctrines (Kaminski, 2023), as the new law for "Risky Agents without Intentions" (Ayres & Balkin, 2024).

## 7 HOW TO MAKE AGENCY LAW AS USEFUL ANALOGIES

### 7.1 STATUTORY REGIMES FOR POLYADIC LIABILITY ALLOCATION

Agency law, premised on bilateral human-to-human relationships, provides little guidance when AI agents must navigate conflicting instructions from multiple stakeholders. Consider existing regulatory models that already address polyadic governance structures. The Investment Advisers Act of 1940 imposes fiduciary duties on financial advisors while recognizing that multiple parties (investment managers, broker-dealers, custodians) participate in the advisory relationship, with detailed regulations specifying each actor's boundaries of duties and potential liability (Randall, 1978). Similarly, California's Talent Agencies Act regulates entertainment agents by defining obligations for

agents, personal managers, and production companies, acknowledging that talent representation involves multiple intermediaries with conflicting interests (Smith, 2019).

AI agent governance requires analogous but more granular statutory specificity corresponding to the complex nature of governance. Agency law's general principles of loyalty and care cannot adequately address situations where foundation model providers, fine-tuners, deployers, and users each impose different constraints on agent behavior. When an AI agent produces harm, the uncertain nature of how training decisions, safety constraints, and deployment configurations interact makes fault-based liability allocation impractical (Cabral, 2020; Cheong, 2025). Proving which specific actor's decision caused the harm becomes prohibitively difficult given the opacity of model and the distributed nature of control.

A more effective solution requires legal architectures that impose ex-ante duties on multiple actors in the supply chain and distribute liability without requiring strict proof of fault. This allocation should consider evidentiary access to information, control over risk at different stages, and policy considerations balancing innovation and safety. The EU AI Act and revised Product Liability Directive exemplify this model (EU, 2024a). The Act imposes differentiated obligations on providers, deployers, importers, and distributors. The revised Product Liability Directive establishes strict liability for defective products, creates rebuttable presumptions to ease plaintiff burdens of proof, and supports joint and several liability (EU, 2024b). These instruments address polyadic governance by specifying actor-specific duties ex ante and enabling burden-shifting ex post.

## 7.2 TECHNICAL GOVERNANCE MECHANISMS

Technical mechanisms are necessary to operationalize accountability across polyadic governance structures. We outline a minimal stack that indicates feasible directions rather than prescribing full engineering specifications. First, provenance-aware documentation should record which actor shaped system behavior at each stage. Telemetry frameworks such as OpenTelemetry's semantic conventions already provide foundations for standardized logging across reasoning steps and tool calls (Young & Parker, 2024). Second, governance-chain logging should enable auditable reconstruction from training through deployment. This is essential when provider-imposed rules override user instructions and helps limit responsibility shifting observed in regulatory circumvention.

Third, evaluation frameworks should include metrics suited to polyadic systems. Goal-consistency tests examine whether agents satisfy constraints from multiple stakeholders. Tools such as Microsoft's PyRIT (Munoz et al., 2024) and evaluations by NIST ARIA (Schwartz et al., 2024) illustrate emerging methods, but standardized third-party protocols are necessary to avoid selective testing. Fourth, mechanisms for documenting conflicts should transparently record when provider rules or incentives override user goals. Because platform agents centralize control with providers, structural conflicts of interest are unavoidable. One promising model is the use of agent advocates—independent intermediaries that represent user interests in configuring, monitoring, and auditing agents (Kapoor et al., 2025). Proposals such as California's SB 813 illustrate how these features can be institutionalized through multi-stakeholder governance (Carlson, 2025).

## 8 CONCLUSION

AI agents are rapidly moving from experimental tools to embedded infrastructure in both consumer and enterprise settings. As they take on increasingly autonomous, judgment-like tasks, questions of **Agency**, **Loyalty**, and **Accountability** become unavoidable. Yet today's agents operate through fragmented layers of control—developers, providers, and users each shaping behavior in ways that prevent undivided loyalty or clear responsibility. Existing legal frameworks risk creating only the illusion of faithful agents, encouraging users to rely on them while leaving providers insulated from liability. In this paper, we highlight the structural differences between AI systems and human Agents, showing why familiar doctrines of agency law, while tempting, cannot be transplanted without distortion. By surfacing the limits of current approaches, we reframe debates about AI governance and provide a foundation for developing new institutional, technical, and legal mechanisms. Addressing these challenges will be essential to ensure that as AI agents become more deeply integrated into daily life, they operate under structures that distribute control and responsibility in ways that are both fair and trustworthy.

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

# A APPENDIX

## A.1 USE OF LARGE LANGUAGE MODELS

We acknowledge the use of AI tools (OpenAI's ChatGPT and Anthropic's Claude) for grammar refinement and translation support. All substantive arguments and analyses are the authors' own.

## A.2 LEGAL DISCUSSIONS ON AGENCY IN DIGITAL SERVICES

Scholars have explored whether principles of agency and fiduciary duty could provide governance models for digital platforms, particularly social media companies.

Jack Balkin introduced the concept of "information fiduciaries" in 2015, arguing that because technology companies collect, store, and use vast amounts of personal data, they should be subject to ongoing fiduciary-like duties similar to financial advisors handling clients' assets Balkin (2015). His proposal was partly pragmatic: fiduciary duties, rooted in common law, might raise fewer free-speech concerns than regulatory regimes modeled on the GDPR (e.g., the "right to be forgotten").

Building on this idea, Neil Richards and Woodrow Hartzog expanded the notion of a **duty of loyalty** for digital platforms Richards & Hartzog (2021); Hartzog & Richards (2021). They argued that fiduciary framing better addresses the power asymmetries between platforms and users than the widely discredited "notice-and-consent" model. James Grimmelmann similarly suggested that search engines might be understood as "trusted advisors," with obligations to provide results that genuinely serve user needs Grimmelmann (2013).

Despite these theoretical developments, the analogy between fiduciaries and online platforms has been contested. Lina Khan and David Pozen dismissed fiduciary duties in this context as largely ineffective, unable to resolve conflicts both among users and between platforms' obligations to users and to shareholders Khan & Pozen (2019). Claudia Haupt also argued that the lawyer-client or doctor-patient fiduciary model is ill-suited to platforms that manage information flows at scale rather than provide individualized counsel Haupt (2020). She suggested that the trustee–beneficiary analogy may be a better fit.

Importantly, these debates remained mostly academic. Unlike traditional fiduciaries or agents, social media platforms are not typically perceived as acting "on behalf of" users. Instead, they serve multiple users simultaneously, often balancing conflicting interests—for instance, moderating harmful content while preserving free expression. As a result, while the discourse on "information fiduciaries" generated valuable normative insights, it has not translated into legal or institutional practice. Richards and Hartzog have continued to advocate for legislating duties such as loyalty Richards et al. (2023); Hartzog & Richards (2022), but the conversation largely stalled until the recent rise of AI agents reignited questions about agency in digital contexts.

## A.3 DELEGATION, INTERACTIVITY, AND AUTONOMY ON DIGITAL SERVICES

## A.4 COURTS' REASONING ON RESPONDEAT SUPERIOR

Table 3: Comparison of digital services, AI agents, and human agents

| Property | Wikipedia | Amazon | AI agents | Human Agents |
|---|---|---|---|---|
| **Delegation** | Users retrieve information directly; no task execution. | Users specify items and transactions; platform executes predefined workflows. | Users delegate goals (*"book me a flight"*); agent decomposes into subtasks, applies constraints, executes. | Users delegate outcomes broadly; human agent interprets intent, applies judgment, handles exceptions. |
| **Interactivity** | Static interaction: query and read results; no context across sessions. | Structured interactions: browse, purchase, track; limited conversational support. | Dynamic, multi-modal: natural language conversations, API calls, negotiation with third parties, memory of context. | Rich, adaptive: nuanced communication, persuasion, empathy, social intelligence. |
| **Autonomy** | None: system is passive, user-driven. | Low: limited automation (recommendations, order tracking) but not proactive. | Medium–High: initiative (event triggers), adaptation (plan revision), persistence (long-running workflows). | High: can self-initiate, deeply adapt, sustain long-term projects, improvise under uncertainty. |

Table 4: Scope of Employment Analysis: Employer Liability to Third Parties

| Employee Conduct | Employer Liable? | Rationale |
|---|---|---|
| Employee makes intentional misrepresentations to prospective customers to induce purchases | Yes | Making statements to customers is within assigned job duties Quick v. Peoples Bank (1993) |
| Employee drives negligently while performing delivery duties | Yes | Driving is part of assigned task; negligence is foreseeable Hinman v. Westinghouse Electric Co. (1970) |
| Employee slams trays during heated customer complaint, injuring customer | Yes | Emotionally-driven conduct while performing assigned work (handling complaints) Lee v. United States (2001) |
| Truck driver chats on cell phone, becomes distracted, and causes accident | No | Personal phone call is a non-work-related independent course of action Haybeck v. Prodigy Servs. Co. (1996) |
| Irate driver shoots another driver while driving company truck | No | Extreme violence exceeds any reasonable scope of employment Monty v. Or-landi (1959) |
| Inebriated seaman turns valves on drydock wall, causing flooding and ship damage | Yes | Foreseeable risk of seamen's conduct; act not entirely due to personal life Bushey v. United States (1968) |

