# OpenReview forum: "Agents Aren't Agents: the Agency, Loyalty and Accountability Problems of AI agents"
_ICLR.cc/2026/Conference — Submitted to ICLR 2026_

### Official Review · Reviewer_HBMk · 2025-10-28

**Soundness:** 3
**Presentation:** 3
**Contribution:** 2
**Rating:** 4
**Confidence:** 3

**Summary:**

This position paper contends that today’s so‑called “AI agnts” are not legal Agents and cannot be treated as such without distortion. The authors claim that the analogy to human agency law breaks for three reasons they term the Agency, Loyalty, and Accountability problems.

1st: Agency (who is the principal/agent?) AI systems are governed “polyadically (by trainers, hosts, tool/wrapper develoers, and end‑users). 2nd: Loyalty (can an AI reliably act in the principal’s best interests?) Even absent self‑interest, models routinely display disloyal behavior through instruction‑following brittleness, hallucination, non‑determinism, and provider conflicts of interest. 3rd, Accountability (who is liable when things go wrong?) Traditional mechanisms (e.g., fiduciary liability or respondeat superior) do not map cleanly onto AI systems.

The paper’s contribution is to re‑center discussions of “AI agents” around legal agency’s core doctrines, showing why doctrinal transpants are dangerous, and to surface gaps that invite new technical and legal research. The work is intentionally diagnostic rather than prescriptive (Abstract, p. 1; Conclusion, p. 9).

**Strengths:**

- Clear reframing of the “AI agent” metaphor. The paper shows, with diagrams, how today’s systems lack the structural features assumed by agency law (Figure 2, p. 5; Figure 3, p. 5). This helps ML practitioners avoid over‑reliance on legal analogies.

- Useful taxonomy of misconceptions and duties. Table 1 (p. 2) and the summary of fiduciary/accountability principles (Table 2, p. 4) give readers a map of doctrine and where it fails to transfer.

- The discussion of instruction brittleness, hallucination, and stochasticity as “disloyalty” in a fiduciary sense (Sec. 5.1, pp. 6–7) is insightful for an ML audience.

- Candid scope and positioning. The paper states it is a position piece aimed at surfacing issues and stimulating research rather than settling doctrine.

**Weaknesses:**

Major concerns :
The analysis leans heavily on the U.S. Restatement and state doctrines (pp. 4–9) and offers little on the EU AI Act’s liability interfaces or product‑liability modernization, nor on civil‑law analogs of agency.

Under‑argued leap from “AI cannot be Subagents” to “providers should assume 100% responsibility.” Section 4.2 (pp. 6–7) asserts that AI agents cannot be subagents and concludes the only plausible option is full provider responsibility. The normative basis and feasibility are not fully defended. A more detailed allocation model (rebuttable presumptions, strict liability bands by capability/risk) would strengthen the claim.

Section 6.2 (pp. 8–9) argues that respondeat superior is a poor fit largely because models lack personal motives. Courts often ask about foreseeability and scope of assigned tasks; these could, in principle, capture many AI behaviors. Engaging closely with how “scope of employment” could be reinterpreted for technical artifacts, and with edge cases like autonomous prevention/mitigation features, would improve soundness.

The paper largely stops at “structural mismatch.” Readers would benefit from concrete, technically actionable implications (e.g., logging standards for allocatable causation; loyalty tests/benchmarks; verifiable delegation protocols). The brief references to safety guardrails and provider discretion (pp. 6–7) could be expanded into design patterns.

Minor concerns:
Ambiguity in the five cases (Figure 3). The mapping of the “Cursor updates your blog” example to Case 5 vs. the airline booking example to Case 4 is terse (p. 5) and may confuse readers about when no agency vs. provider‑as‑principal applies. A small decision tree would help.

**Questions:**

Comparative law: How would your Agency/Loyalty/Accountability framing change under the EU AI Act and proposed Product Liability Directive revisions? Can you sketch how “polyadic governance” interacts with strict liability proposals in the EU?

Allocation model: Instead of 100% provider responsibility (Sec. 4.2), would you endorse a rebuttable‑presumption model that places initial liability on the deployer/provider but allows upstream indemnities conditioned on demonstrable controls?

Benchmarks for “loyalty.” Could the ML community help with loyalty benchmarks (multi‑constraint compliance under safety overrides, conflict‑of‑interest stress tests)? What measurable targets would meaningfully inform legal duties of care ?

---

> ### Author Response · Authors · 2025-11-23
> **Response 1 to Reviewer HBMk**
>
> We appreciate the reviewer’s concern that the earlier version placed greater emphasis on diagnosis than implementation. We have revised the manuscript to clarify how the conceptual framework translates into practice.
>
> ## 1. Lack of Comparative Law Analysis
>
> The reviewer observes that the analysis relies heavily on U.S. common law doctrines without addressing EU AI Act liability interfaces or civil law analogs, and suggests that the paper would benefit from a more detailed allocation model that incorporates rebuttable presumptions and strict liability bands by capability or risk.
>
> Thank you for this thoughtful review. We appreciate the opportunity to clarify our comparative law perspective and allocation model.
>
> ### Agency Law as a Common Law Framework
> Agency law and fiduciary duties are deeply rooted in common law traditions. Discussing civil law analogies requires substantial groundwork that this paper does not attempt. As Gelter and Hellerich note, civil law systems produce practically equivalent outcomes through other legal concepts without relying on separate fiduciary categories, making translation difficult across legal languages [1]. This reflects structural features of common law systems, including the historical division between law and equity, greater emphasis on contractual autonomy, and less reliance on regulatory regimes. Judge-made fiduciary principles override party-made contracts in specific relationships. When professions become subject to public regulation, the need for distinct fiduciary principles as organizing concepts diminishes [2].
>
> Consider how fiduciary professions in the United States face different regulatory intensities. Financial advisors now operate under extensive federal regulation through successive reforms, and their governance framework increasingly resembles field-specific consumer protection law, paralleling the EU AI Act model. Lawyers face state-specific regulation, though legislative, judicial, and bar-association authority is distributed differently across jurisdictions. Talent agents receive specific regulation only in states such as California. These variations reflect governance judgments about when executive agencies should monitor and enforce private contracts rather than relying on judicial dispute resolution.
>
> ### The Regulatory Design Debate
> Proposals to apply fiduciary principles to digital services have sometimes been criticized as attempts to forestall comprehensive regulation [2]. We partly share this concern. As Appendix A.2 notes, Jack Balkin advocated fiduciary duties for digital platforms in 2015, emphasizing practical benefits: the doctrine’s acceptance in common law avoids First Amendment conflicts and skepticism toward comprehensive privacy regulation [3]. Lina Khan and David Pozen responded that this approach “invites an enervating complacency about issues of structural power and a premature abandonment of more robust visions of public regulation” [2]. They argued that efforts should focus on pro-competition rules and consumer protections rather than flexible, relational categories that promise bipartisan appeal without robust regulatory intervention.
>
> We agree that agency law or fiduciary law can appear as de-regulatory approaches. These doctrines emerged from judge-adjudicated dispute resolution without executive agency involvement. Legislating an ambiguous mandate such as “agents must be loyal to users” offers no meaningful guidance on when developer-imposed rules should yield to user requests. Providing such guidance requires statutory regulation. Once statutory detail increases (as with financial advisors), the regime becomes functionally indistinguishable from field-specific consumer protection law.
>
> ### Our Position on AI Agent Governance
> As Section 2 explains, AI agents are more relational than the digital platforms Khan and Pozen examined, making agency analogies partially useful for diagnosis. Yet financial advisors have received escalating public regulation precisely because flexibility and ad-hoc ex-post adjudication proved insufficient. We believe AI agents require a comparable evolution toward consumer protection approaches or duty-based rules such as those embodied in the EU AI Act.
>
> Promoting agency law or fiduciary duties as flexible, case-specific substitutes for comprehensive regulation offers false promises. AI agents can generate more significant and unpredictable harms than financial advisors. Meaningful loyalty duties require addressing AI agent corporate governance structures, actor-specific ex-ante rules, and preventive auditing and monitoring rather than case-specific judicial discovery after harm occurs. These conditions support a governance model that cannot rely solely on relational doctrines and instead depends on coordinated regulatory obligations that operate across the entire lifecycle of AI systems, from design and training to deployment and ongoing oversight.

---

> ### Author Response · Authors · 2025-11-23
> **Response 2 to Reviewer HBMk**
>
> ### Clarifications and Revisions
> Sections 5.2 and 6.3 already indicate that agency law principles fail to guide the balancing of interests, allowing agents “expansive opportunities for interest seeking at the expense of their Principals” through their “discretion in rule-making and execution,” thereby necessitating statutory regulation. Section 6.3 further notes: “These difficulties have prompted proposals to reallocate burdens of proof, to adopt rebuttable presumptions, or to move toward strict liability regimes.”
>
> We did not explicitly endorse EU AI Act approaches over common law frameworks because a deeper comparative diagnosis was necessary. However, we recognize that readers expect prescriptive guidance, and our diagnosis aligns with ex-ante regulation and the revised Product Liability Directive’s approach to reducing plaintiff burdens of proof. We will therefore add Section 7 to address allocation models and EU regulatory interfaces more explicitly.
>
> The combined operation of the AI Act and the New Product Liability Directive forms a dual structure that pairs ex-ante duties of design, monitoring, documentation, and update with ex-post strict or burden-shifting liability. This structure aligns with the polyadic governance of AI agents and reinforces the view that managing AI-related risks requires systemic oversight rather than flexible, case-specific common-law doctrines.
>
> The New Product Liability Directive explicitly recognizes software and AI as “products,” ending debates about whether AI-enabled services should be covered; extends liability to software developers, fulfillment service providers, authorized representatives, distributors, and, in limited circumstances, platforms; and introduces presumptions that ease burdens of proof in cases involving scientific or technical complexity, directly acknowledging the opaque and distributed nature of AI-related harms.
>
> ### Proposed Addition of Section 7.1
> We propose the following addition to Section 7.1 in the revised manuscript.
>
> ---
>
> > ## **Section 7. AI Agent Governance Beyond Agency Law: Legal Regimes and Technical Design**
> >
> > ### **7.1 Statutory Regimes for Polyadic Liability Allocation**
> >
> > Traditional agency law, premised on bilateral human-to-human relationships, provides little guidance when AI agents must navigate conflicting instructions from multiple stakeholders. Consider existing regulatory models that already address polyadic governance structures. The Investment Advisers Act of 1940 imposes fiduciary duties on financial advisors while recognizing that multiple parties (investment managers, broker-dealers, custodians) participate in the advisory relationship, with detailed regulations specifying each actor's boundaries of duties and potential liability.
> >
> > Similarly, California's Talent Agencies Act regulates entertainment agents by defining specific obligations for agents, personal managers, and production companies, acknowledging that talent representation involves multiple intermediaries with potentially conflicting interests.
> >
> > AI agent governance requires analogous but more granular statutory specificity corresponding to the complex nature of governance. Agency law's general principles of loyalty and care cannot adequately address situations where foundation model providers, fine-tuners, deployers, and users each impose different constraints on agent behavior.
> >
> > When an AI agent produces harm, the uncertain nature of how training decisions, safety constraints, and deployment configurations interact makes fault-based liability allocation impractical. Proving which specific actor's decision caused the harm becomes prohibitively difficult given the opacity of model behavior and the distributed nature of control.
> >
> > A more effective solution requires comprehensive legal architectures that impose ex-ante duties on multiple actors in the supply chain and distribute liability without requiring strict proof of fault. This allocation should consider three dimensions:
> > - evidentiary access to information,
> > - control over risk at different lifecycle stages,
> > - and policy considerations balancing innovation and safety.
> >
> > The EU AI Act and revised Product Liability Directive exemplify this model. The AI Act imposes differentiated obligations on providers, deployers, importers, and distributors. The revised Product Liability Directive establishes strict liability for defective products, creates rebuttable presumptions to ease plaintiff burdens of proof, and supports joint and several liability. Together, these instruments address polyadic governance by specifying actor-specific duties ex ante and enabling burden-shifting ex post.
> >
> > ---

---

> ### Author Response · Authors · 2025-11-23
> **Response 3 to Reviewer HBMk**
>
> ## 2. Reinterpretation of *Respondeat Superior* Could Be Helpful
>
> We appreciate the reviewer’s observation that courts’ inquiries into foreseeability and the scope of assigned tasks could, in principle, capture many AI behaviors, and that closer engagement with how “scope of employment” might be reinterpreted for technical artifacts would improve soundness.
>
> Upon closer examination of this critique, we recognize that our analysis requires refinement. Section 6.2 argues that *respondeat superior* is a poor fit largely because AI agents lack personal motives that typically mark conduct as outside the scope of employment. While we maintain that *respondeat superior* cannot adequately address AI agent liability without substantial modification, the reviewer correctly identifies that certain conceptual elements of the doctrine remain relevant. Specifically, the principles of imposing liability without requiring proof of fault (as a form of risk allocation) and using foreseeability as an evaluative criterion can inform AI agent liability frameworks. However, these principles must be recontextualized within entirely new regimes rather than applied through *respondeat superior*’s employment-based structure.
>
> Foreseeability remains crucial for AI agent liability, and the EU’s revised Product Liability Directive illustrates this by incorporating “reasonably foreseeable use of the product” (Art. 7(2)(b)) and “reasonably foreseeable effects of other products” (Art. 7(2)(d)) into its determination of defectiveness. Importantly, foreseeability in this context derives from general tort principles rather than *respondeat superior*’s employment relationship framework. Similarly, non-fault risk allocation can apply to AI systems, but *respondeat superior* assumes bilateral employment relationships where employees bear primary liability and employers assume vicarious liability. AI agents cannot bear legal responsibility directly. If providers avoid liability by invoking unforeseeability, third parties face a liability gap.
>
> Moreover, *respondeat superior*’s assumptions about employer control do not translate to AI agents. Traditional applications presume employers can exercise meaningful control over employee conduct through hiring, training, supervision, and discipline. By contrast, AI agent behavior is shaped by multiple actors—foundation model trainers, safety teams, providers, fine-tuners, deployers—none of whom exercises comprehensive control. The doctrine’s foreseeability inquiry becomes unstable when applied to distributed systems where “control” fragments across several parties.
>
> The nature of “unforeseeability” also differs. Under *respondeat superior*, unforeseeability typically refers to personal deviation by an employee. For AI agents, problematic behaviors often arise not from personal deviation but from structural limits and distributed control. When an AI agent generates harmful output, determining whether the harm occurred “within scope” becomes analytically ambiguous. Did it stem from assigned functionality (suggesting deployer responsibility), from model design choices (suggesting provider involvement), or from inadequate safeguards (suggesting deployer negligence)? The doctrine provides little guidance for allocating responsibility among multiple actors.
>
> In conclusion, while *respondeat superior* offers useful ideas—foreseeability and vicarious liability without fault—extending the doctrine itself is insufficient. Properly allocating risk and responsibility requires liability regimes tailored to polyadic governance structures. The EU’s revised Product Liability Directive provides such a model. We will revise Section 6.2 to reflect this more nuanced analysis.
>
> ---
>
> ## 3. Poorly Supported Claim That “AI Cannot Be Subagents”
>
> We appreciate the reviewer’s concern that Section 4.2 makes an under-argued leap from “AI cannot be Subagents” to “providers should assume 100% responsibility,” and that a more detailed allocation model would clarify the argument.
>
> Our intention was narrower than implying that providers should normatively bear full responsibility. Our point was doctrinal: the Subagent concept cannot function as a liability-limiting mechanism for AI. Under agency law, Subagents must (1) be liable parties capable of bearing responsibility and (2) prioritize the Principal’s interests over the Agent’s interests. AI agents meet neither requirement. They lack legal personhood, cannot bear liability, and cannot reliably prioritize Principal instructions when provider-imposed safety rules override user directives.
>
> Because AI agents cannot serve as Subagents under agency law, the Subagent framework collapses, leaving AI service providers as direct Agents under traditional logic. This doctrinal limitation means that attempts to limit provider liability through Subagent theories are unavailable.

---

> ### Author Response · Authors · 2025-11-23
> **Response 4 to Reviewer HBMk**
>
> This does not mean providers should normatively bear full responsibility. Allocation models involving rebuttable presumptions, joint liability with indemnity, or capability-based risk tiers may be more appropriate. Any such allocation must be grounded in statutory regimes or alternative legal theories rather than Subagent doctrine. This reinforces the paper’s broader argument that AI agent governance requires new liability regimes tailored to polyadic control structures.
>
> We propose to revise Section 4.2 along the following lines:
>
> > “However, this doctrinal route is unavailable. Agency law expects Subagents to (1) be personally liable to the Principal and (2) prioritize the Principal’s interests over the Agent’s interests (American Law Institute, 2006, § 3.15). AI agents meet neither condition. They cannot be held liable without legal personhood, assets, or capacity for legal consequences, and they cannot consistently prioritize the Principal’s instructions since developer-imposed safety constraints override user directives. Because AI agents cannot serve as liable Subagents, the Subagent framework collapses, leaving AI service providers as direct Agents under *respondeat superior* logic.
> >
> > This does not mean providers should normatively bear full responsibility. Alternative allocation models such as rebuttable presumptions, joint liability with indemnity rights, or capability-based risk allocation may be preferable. Any allocation of responsibility between providers and deployers must be grounded in alternative legal theories, reinforcing the need for liability regimes tailored to polyadic control structures, as exemplified by the EU’s revised Product Liability Directive.”
>
> ---
>
> ## 4. Technical Mitigation Solutions
>
> We appreciate the reviewer’s observation that the paper largely diagnoses structural mismatch but would benefit from concrete, technically actionable implications. We agree that prescriptive guidance strengthens the contribution.
>
> Readers asked about logging standards for allocatable causation, loyalty benchmarks, and verifiable delegation protocols. The reviewer also inquires whether the ML community could help develop loyalty benchmarks—such as multi-constraint compliance under safety overrides or conflict-of-interest stress tests—and what measurable targets could inform duties of care.
>
> We agree and propose the following addition as Section 7.2.
>
> > ### **7.2 Technical Governance Mechanisms**
> >
> > While statutory regimes define the legal foundation for responsibility, technical and institutional mechanisms are necessary to operationalize accountability across polyadic governance structures. We outline a minimal accountability stack that indicates feasible directions rather than prescribing full engineering specifications.
> >
> > **First**, provenance-aware documentation should record which actor shaped system behavior at each stage. Telemetry frameworks such as OpenTelemetry’s emerging semantic conventions already provide foundations for standardized logging across reasoning steps and tool calls [4].
> >
> > **Second**, governance-chain logging should enable auditable reconstruction from training through deployment. This is essential when provider-imposed rules override user instructions and helps limit responsibility shifting observed in regulatory circumvention.
> >
> > **Third**, evaluation frameworks should include metrics suited to polyadic systems. Goal-consistency tests examine whether agents satisfy constraints from multiple stakeholders. Partner-steering assessments detect systematic preference for provider interests. Run-to-run variance captures non-deterministic behavior. Tools such as Microsoft’s PyRIT [5] and evaluations by NIST ARIA [6] illustrate emerging methods, but standardized third-party protocols are necessary to avoid selective testing.
> >
> > **Fourth**, mechanisms for documenting conflicts should transparently record when provider rules or incentives override user goals. Because platform-based agents centralize control with providers, structural conflicts of interest are unavoidable. One promising complementary model is the use of **agent advocates**—independent intermediaries that represent user interests in configuring, monitoring, and auditing agents [7]. They can coexist with open-source infrastructures that distribute governance power and reduce reliance on proprietary safety layers.
> >
> > These mechanisms gain legal force through documentation duties, certification conditions, and rules assigning evidentiary weight to logs and provenance records. Proposals such as California’s SB 813 illustrate how these features can be institutionalized through multi-stakeholder governance [8]. Integrating technical documentation with legal standards ensures that accountability does not depend solely on ex-post reconstruction.
> >
> > Detailed engineering specifications remain important future work beyond the scope of this paper.
>
> ---

---

> ### Author Response · Authors · 2025-11-23
> **Response 5 to Reviewer HBMk**
>
> ## 5. Clarifications of Figure 3
>
> We appreciate the reviewer’s observation regarding potential ambiguity in distinguishing Case 4 from Case 5 in Figure 3. Our use of the phrase “at best” for the airline-booking example was intended to signal that, although the AI performs actions aligned with the user’s interests, it still operates in its own name when interacting with third parties. This leaves the user in the position of a third party rather than creating an agency relationship.
>
> In contrast, the “Cursor updates your blog” example belongs in Case 5 because the AI acts directly under the user’s identity and does not form independent relationships with third parties. This removes the structural basis for an agency relationship in the legal sense. We agree that this distinction is subtle and may remain confusing without an explicit decision procedure.
>
> To resolve this, we introduce the following decision tree, which classifies any AI system into one of the five cases. We plan to include this figure in the main text or in an appendix depending on space.
>
> > **Figure X. Decision Tree for Classifying AI Agent Relationships**
> > Five sequential questions determine which of five agency configurations applies:
> > (Q1) whether the AI affects third parties;
> > (Q2) on whose behalf the AI acts;
> > (Q3) whether user approval is required for each action;
> > (Q4) who controls the AI’s goals and constraints; and
> > (Q5) whether users can override provider constraints.
> > Most current systems fall into Case 3 (provider as agent), Case 4 (user as third party), or Case 5 (no agency), leaving users without the protections that traditional agency law would provide in Cases 1 or 2.
> >
> > ---
> >
> > **Q1. Does the AI take actions affecting third parties?**
> > • **No** → **Case 5** (no agency)
> > &nbsp;&nbsp;Example: ChatGPT used for research
> > • **Yes** → Q2
> >
> > ---
> >
> > **Q2. On behalf of whom does the AI act?**
> > • **Provider / Platform** → **Case 4**
> > &nbsp;&nbsp;Example: Airline booking agent (User = third party)
> > • **User** → Q3
> >
> > ---
> >
> > **Q3. Must the user approve each action?**
> > • **Yes** → **Case 5** (AI as a tool)
> > &nbsp;&nbsp;Example: Cursor without automated push
> > • **No** → Q4
> >
> > ---
> >
> > **Q4. Who controls the AI’s goals and constraints?**
> > • **User** → **Case 1**
> > &nbsp;&nbsp;Example: Agent advocates (User = Principal)
> > • **Provider** → Q5
> >
> > ---
> >
> > **Q5. Can the user override the provider’s constraints?**
> > • **Yes** → **Case 2**
> > &nbsp;&nbsp;Example: Investment advisor (AI as Subagent)
> > • **No** → **Case 3**
> > &nbsp;&nbsp;Example: LitAI agency (Provider as sole Agent)
>
>
> We believe the revised manuscript resolves the concerns outlined in the review. Again, thank you for the careful reading and thoughtful suggestions.
>
>
>
> ---
> **References**
>
> [1] Gelter, Martin, and Geneviève Helleringer. *Fiduciary Principles in European Civil Law Systems.* In Evan J. Criddle, Paul B. Miller, and Robert H. Sitkoff (eds.), **The Oxford Handbook of Fiduciary Law**. Oxford University Press, 2019.
> https://doi.org/10.1093/oxfordhb/9780190634100.013.32
>
> [2] Khan, Lina M., and David E. Pozen. “A Skeptical View of Information Fiduciaries.” *Harvard Law Review* 133, no. 2 (2019): 497–541.
>
> [3] Balkin, Jack M. “Information Fiduciaries and the First Amendment.” *UC Davis Law Review* 49 (2015): 1183–1234.
>
> [4] Young, Ted, and Austin Parker. *Learning OpenTelemetry*. Sebastopol, CA: O’Reilly Media, 2024.
>
> [5] Munoz, Gary D. Lopez, Amanda J. Minnich, Roman Lutz, Richard Lundeen, Raja Sekhar Rao Dheekonda,
>     Nina Chikanov, Bolor-Erdene Jagdagdorj et al.
>     "PyRIT: A Framework for Security Risk Identification and Red Teaming in Generative AI Systems."
>     arXiv preprint arXiv:2410.02828 (2024).
>
> [6] Schwartz, Reva, Gabriella Waters, Razvan Amironesei, Craig Greenberg, Jon Fiscus, Patrick Hall,
>     Anya Jones et al.
>     "The Assessing Risks and Impacts of AI (ARIA) Program Evaluation Design Document." (2024).
>     https://ai-challenges.nist.gov/aria/docs/ARIA_Program_Companion_Document_Dec20.pdf
>
> [7] Kapoor, Sayash, Noam Kolt, and Seth Lazar.
>     "Position: Build Agent Advocates, Not Platform Agents."
>     In *Forty-Second International Conference on Machine Learning* (ICML) Position Paper Track, 2025.
>
> [8] Cal. S.B. 813, 2025–2026 Reg. Sess. (as amended May 23, 2025).

---

### Official Review · Reviewer_AwkX · 2025-11-01

**Soundness:** 3
**Presentation:** 3
**Contribution:** 3
**Rating:** 6
**Confidence:** 2

**Summary:**

This position paper investigates why current AI agents don’t fit precisely into the legal category of human agents. The paper analyzes three issues: Agency, Loyalty, and Accountability.

**Strengths:**

The paper is interdisciplinary. It bridges AI technical problems and legal theory. This is very relevant for regulators, users, and platform owners. It is also timely as 2025 is called the “year of AI agents”.
The paper reveals concrete risks that can guide policy and technical approaches.

**Weaknesses:**

The paper discusses mostly the common law in the united states but it is not clear how this analysis carries over to other regions.
Although the paper investigates the problems clearly, it could benefit from proposing concrete solutions.

**Questions:**

Have you considered any empirical study of commercial agent ToS, logging practices, or reported incidents to illustrate the “avoision” patterns you describe?

---

> ### Author Response · Authors · 2025-11-23
> **Response 1 to Reviewer AwkW**
>
> We appreciate the reviewer’s concern that the earlier version placed greater emphasis on diagnosis than implementation. We have revised the manuscript to clarify how the conceptual framework translates into practice.
>
> We thank the reviewer for the thoughtful and constructive review. We particularly appreciate the reviewer’s recognition of the paper's interdisciplinary contribution, its timeliness in the "year of AI agents," and your observation that our analysis reveals concrete risks that can guide policy and technical approaches. This assessment affirms the value of bridging AI technical problems and legal theory for regulators, users, and platform owners.
>
> ## Comparative Law Scope
> The reviewer notes that the paper focuses on U.S. common law and asks how the analysis carries over to other regions. This is an important question. While agency law and fiduciary duties are deeply rooted in common law traditions, we examine how the principles of foreseeability-based liability and non-fault risk allocation can inform AI agent governance more broadly. Our analysis of the EU AI Act and revised Product Liability Directive demonstrates this translation, as described in greater detail in the response to Reviewer HBMk. These instruments address polyadic governance by establishing strict liability without requiring proof of employment-like relationships, creating rebuttable presumptions to ease plaintiff burdens, and enabling joint and several liability among multiple economic operators. Therefore, we propose the addition of Section 7.1, which discusses how these EU frameworks exemplify legal regimes that impose ex-ante duties on multiple actors while enabling liability allocation ex post based on evidentiary, control, and policy considerations rather than bilateral agency relationships.
>
> ## Concrete Solutions
> The reviewer suggests the paper would benefit from proposing concrete solutions beyond problem diagnosis. We agree and will add Section 7 to address this concern. Section 7.1 discusses statutory frameworks for polyadic liability allocation, examining how the Investment Advisers Act of 1940 and California's Talent Agencies Act provide precedents for multi-party governance structures, and how the EU AI Act and revised Product Liability Directive extend these models to AI systems. The analysis considers three allocation dimensions: evidentiary considerations (which actors have superior access to information), control considerations (which actors can prevent specific harms), and policy considerations (balancing innovation incentives against safety imperatives). Section 7.2 proposes technical governance mechanisms, including standardized observability protocols through OpenTelemetry, independent evaluation frameworks to prevent selective testing, and design patterns for auditable logging and transparency. These technical mechanisms can be institutionalized through multiple legal pathways: as mandatory duties, as conditions for certification by regulatory bodies like California's proposed Multistakeholder Regulatory Organization, or by specifying their evidentiary weight for liability allocation.
>
> ## Empirical Evidence
> We appreciate the reviewer's suggestion to incorporate empirical analysis of commercial agent terms of service, logging practices, and reported incidents. While a systematic empirical study of these practices lies beyond the scope of this primarily doctrinal analysis, we acknowledge that such evidence would strengthen the practical grounding of our framework.
>
> Current commercial AI agent deployments do reveal patterns consistent with our analysis of polyadic governance challenges. Major providers' terms of service for agent platforms typically disclaim liability for agent outputs while simultaneously restricting how deployers can modify safety constraints, creating the divided loyalty problem we identify in Section 5.2. However, the opacity of logging practices and the nascent state of the commercial agent market limit what can be systematically documented at present. Most providers do not publicly disclose whether they maintain auditable records of which actor (provider, deployer, or user) imposed specific constraints that shaped agent behavior in particular instances.
>
> The technical mechanisms we propose in Section 7.2—including standardized logging with data provenance tracking and independent evaluation protocols—are designed precisely to address this current lack of transparency and accountability infrastructure. These mechanisms would make the allocation challenges we identify empirically tractable by creating verifiable records of multi-actor decision-making. We agree that future empirical work examining how these technical governance mechanisms perform in practice, and how commercial practices evolve in response to emerging liability frameworks, would provide valuable insights for refining AI agent governance regimes.

---

> ### Author Response · Authors · 2025-11-23
> **Response 2 to Reviewer AwkX**
>
> The following is the draft of Section 7 we plan to include in the manuscript.
>
> ---
>
> > ## **Section 7. AI Agent Governance Beyond Agency Law: Legal Regimes and Technical Design**
> >
> > ### **7.1 Statutory Regimes for Polyadic Liability Allocation**
> >
> > Traditional agency law, premised on bilateral human-to-human relationships, provides little guidance when AI agents must navigate conflicting instructions from multiple stakeholders. Consider existing regulatory models that already address polyadic governance structures. The Investment Advisers Act of 1940 imposes fiduciary duties on financial advisors while recognizing that multiple parties (investment managers, broker-dealers, custodians) participate in the advisory relationship, with detailed regulations specifying each actor's boundaries of duties and potential liability.
> >
> > Similarly, California's Talent Agencies Act regulates entertainment agents by defining specific obligations for agents, personal managers, and production companies, acknowledging that talent representation involves multiple intermediaries with potentially conflicting interests.
> >
> > AI agent governance requires analogous but more granular statutory specificity corresponding to the complex nature of governance. Agency law's general principles of loyalty and care cannot adequately address situations where foundation model providers, fine-tuners, deployers, and users each impose different constraints on agent behavior.
> >
> > When an AI agent produces harm, the uncertain nature of how training decisions, safety constraints, and deployment configurations interact makes fault-based liability allocation impractical. Proving which specific actor's decision caused the harm becomes prohibitively difficult given the opacity of model behavior and the distributed nature of control.
> >
> > A more effective solution requires comprehensive legal architectures that impose ex-ante duties on multiple actors in the supply chain and distribute liability without requiring strict proof of fault. This allocation should consider three dimensions:
> > - evidentiary access to information,
> > - control over risk at different lifecycle stages,
> > - and policy considerations balancing innovation and safety.
> >
> > The EU AI Act and revised Product Liability Directive exemplify this model. The AI Act imposes differentiated obligations on providers, deployers, importers, and distributors. The revised Product Liability Directive establishes strict liability for defective products, creates rebuttable presumptions to ease plaintiff burdens of proof, and supports joint and several liability. Together, these instruments address polyadic governance by specifying actor-specific duties ex ante and enabling burden-shifting ex post.
> >
> > ---
> >
> > ### **7.2 Technical Governance Mechanisms**
> >
> > While statutory regimes define the legal foundation for responsibility, technical and institutional mechanisms are necessary to operationalize accountability across polyadic governance structures. We outline a minimal accountability stack that indicates feasible directions rather than prescribing full engineering specifications.
> >
> > **First**, provenance-aware documentation should record which actor shaped system behavior at each stage. Telemetry frameworks such as OpenTelemetry’s emerging semantic conventions already provide foundations for standardized logging across reasoning steps and tool calls [1].
> >
> > **Second**, governance-chain logging should enable auditable reconstruction from training through deployment. This is essential when provider-imposed rules override user instructions and helps limit responsibility shifting observed in regulatory circumvention.
> >
> > **Third**, evaluation frameworks should include metrics suited to polyadic systems. Goal-consistency tests examine whether agents satisfy constraints from multiple stakeholders. Partner-steering assessments detect systematic preference for provider interests. Run-to-run variance captures non-deterministic behavior. Tools such as Microsoft’s PyRIT [2] and evaluations by NIST ARIA [3] illustrate emerging methods, but standardized third-party protocols are necessary to avoid selective testing.
> >
> > **Fourth**, mechanisms for documenting conflicts should transparently record when provider rules or incentives override user goals. Because platform-based agents centralize control with providers, structural conflicts of interest are unavoidable. One promising complementary model is the use of **agent advocates**—independent intermediaries that represent user interests in configuring, monitoring, and auditing agents [4]. They can coexist with open-source infrastructures that distribute governance power and reduce reliance on proprietary safety layers.

---

> ### Author Response · Authors · 2025-11-23
> **Response 3 to Reviewer AwkW**
>
> >
> > These mechanisms gain legal force through documentation duties, certification conditions, and rules assigning evidentiary weight to logs and provenance records. Proposals such as California’s SB 813 illustrate how these features can be institutionalized through multi-stakeholder governance [5]. Integrating technical documentation with legal standards ensures that accountability does not depend solely on ex-post reconstruction.
> >
> > Detailed engineering specifications remain important future work beyond the scope of this paper.
>
> We hope these revisions make the contribution clearer and demonstrate how the conceptual framework can guide both policy development and technical design.
>
> ---
> **References**
>
> [1] Young, Ted, and Austin Parker. *Learning OpenTelemetry*. Sebastopol, CA: O’Reilly Media, 2024.
>
> [2] Munoz, Gary D. Lopez, Amanda J. Minnich, Roman Lutz, Richard Lundeen, Raja Sekhar Rao Dheekonda,
>     Nina Chikanov, Bolor-Erdene Jagdagdorj et al.
>     "PyRIT: A Framework for Security Risk Identification and Red Teaming in Generative AI Systems."
>     arXiv preprint arXiv:2410.02828 (2024).
>
> [3] Schwartz, Reva, Gabriella Waters, Razvan Amironesei, Craig Greenberg, Jon Fiscus, Patrick Hall,
>     Anya Jones et al.
>     "The Assessing Risks and Impacts of AI (ARIA) Program Evaluation Design Document." (2024).
>     https://ai-challenges.nist.gov/aria/docs/ARIA_Program_Companion_Document_Dec20.pdf
>
> [4] Kapoor, Sayash, Noam Kolt, and Seth Lazar.
>     "Position: Build Agent Advocates, Not Platform Agents."
>     In *Forty-Second International Conference on Machine Learning* (ICML) Position Paper Track, 2025.
>
> [5] Cal. S.B. 813, 2025–2026 Reg. Sess. (as amended May 23, 2025).

---

### Official Review · Reviewer_EK6J · 2025-11-02

**Soundness:** 2
**Presentation:** 3
**Contribution:** 2
**Rating:** 4
**Confidence:** 3

**Summary:**

The paper claims that current AI agents shouldn’t be treated as legal agents. Since control is split across trainers, hosts, wrappers, and users, these systems can’t offer undivided loyalty, and accountability is not clear. It frames three core problems: Agency (who’s the
principal/agent in this polyadic setup), Loyalty (model anomalies + provider incentives lead to disloyal behavior), and Accountability (classic doctrines like fiduciary duties do not seem to be applicable).

**Strengths:**

The paper tackles a relevant problem. It also has a contribution in how it reframes AI agents through a polyadic-governance lens and the
Agency/Loyalty/Accountability triad, and it states that there is an illegal analogy when considering that today’s systems are anyone’s legal Agent. The paper provides concrete evidence by putting together ML failure modes and provider incentives to specific agency-law doctrines. It’s well written, clear and well-organized.

**Weaknesses:**

The paper does not seem to be mature enough for publication since it does not go much beyond diagnosis. It presents claims about provider conflicts, contractual narrowing, and “divided loyalty”, which are plausible, but discussed just at assertion-level; without any empirical evidence (e.g., a 3-5 platform ToS audit quantifying arbitration clauses, liability caps, training-use terms; a few real and reproducible agent logs or vignettes). Some of the premises are based on human-style “failings” in models; where model deviations should be presumed within scope, placing default liability on providers. The work seems to apply just to US-based scenarios;  and it is not clear to what extent it would be applicable to EU AI Act. The core concept of “polyadic governance” seems to be sound, but it is not clear how to implemented in practice: specify a minimal accountability stack (Authority Manifest, auditable Action Ledger, rebuttable presumptions, and a Loyalty Firewall), as well as concrete metrics (goal-consistency under competing constraints, partner-steering bias, run-to-run variance) and a decision procedure that maps real cases to the categories depicted in Fig. 3.

**Questions:**

No additional questions beyond those outlined in the weaknesses section.

---

> ### Author Response · Authors · 2025-11-23
> **Response 1 to Reviewer EK6J**
>
> We are grateful for the reviewers’ patience. Their detailed feedback prompted a deeper reassessment of several sections, and the revised manuscript reflects that extended reflection.
>
> We appreciate the reviewer’s concern that the earlier version placed greater emphasis on diagnosis than implementation. We have revised the manuscript to clarify how the conceptual framework translates into practice. The proposed Section 7.2 (Please see below) provides a minimal, implementable accountability stack (comprising provenance-aware authority documentation, auditable governance-chain ledgers, evaluation metrics tailored to polyadic structures, and transparent conflict-recording mechanisms) that operationalizes “polyadic governance” without requiring speculative or anthropomorphic assumptions about AI agents.
>
> Regarding the request for empirical evidence (e.g., ToS audits, incident logs, reproducible vignettes), we agree these would be valuable for validating the accountability framework. We clarify in the text that such empirical work is a natural next step but beyond the present paper’s scope. Several of the patterns we highlight, including provider conflicts, contractual narrowing, and responsibility shifting, are well-documented across platform governance scholarship and existing AI accountability reports, and Section 7.2 now connects these documented patterns more explicitly to the proposed governance mechanisms.
>
> To address applicability beyond US doctrine, Section 7.1 also notes how the proposed accountability components can interface with risk-based duty structures, such as those in the EU AI Act and the New Product Liability Directive. These frameworks already rely on traceability, documentation, testing, and conflict-management duties, making our proposal interoperable with EU-style regulatory models.
>
> ---
>
> > ## **Section 7. AI Agent Governance Beyond Agency Law: Legal Regimes and Technical Design**
> >
> > ### **7.1 Statutory Regimes for Polyadic Liability Allocation**
> >
> > Traditional agency law, premised on bilateral human-to-human relationships, provides little guidance when AI agents must navigate conflicting instructions from multiple stakeholders. Consider existing regulatory models that already address polyadic governance structures. The Investment Advisers Act of 1940 imposes fiduciary duties on financial advisors while recognizing that multiple parties (investment managers, broker-dealers, custodians) participate in the advisory relationship, with detailed regulations specifying each actor's boundaries of duties and potential liability.
> >
> > Similarly, California's Talent Agencies Act regulates entertainment agents by defining specific obligations for agents, personal managers, and production companies, acknowledging that talent representation involves multiple intermediaries with potentially conflicting interests.
> >
> > AI agent governance requires analogous but more granular statutory specificity corresponding to the complex nature of governance. Agency law's general principles of loyalty and care cannot adequately address situations where foundation model providers, fine-tuners, deployers, and users each impose different constraints on agent behavior.
> >
> > When an AI agent produces harm, the uncertain nature of how training decisions, safety constraints, and deployment configurations interact makes fault-based liability allocation impractical. Proving which specific actor's decision caused the harm becomes prohibitively difficult given the opacity of model behavior and the distributed nature of control.
> >
> > A more effective solution requires comprehensive legal architectures that impose ex-ante duties on multiple actors in the supply chain and distribute liability without requiring strict proof of fault. This allocation should consider three dimensions:
> > - evidentiary access to information,
> > - control over risk at different lifecycle stages,
> > - and policy considerations balancing innovation and safety.
> >
> > The EU AI Act and revised Product Liability Directive exemplify this model. The AI Act imposes differentiated obligations on providers, deployers, importers, and distributors. The revised Product Liability Directive establishes strict liability for defective products, creates rebuttable presumptions to ease plaintiff burdens of proof, and supports joint and several liability. Together, these instruments address polyadic governance by specifying actor-specific duties ex ante and enabling burden-shifting ex post.
> >
> > ---
> >

---

> ### Author Response · Authors · 2025-11-23
> **Response 2 to Reviewer EK6J**
>
> > ### **7.2 Technical Governance Mechanisms**
> >
> > While statutory regimes define the legal foundation for responsibility, technical and institutional mechanisms are necessary to operationalize accountability across polyadic governance structures. We outline a minimal accountability stack that indicates feasible directions rather than prescribing full engineering specifications.
> >
> > **First**, provenance-aware documentation should record which actor shaped system behavior at each stage. Telemetry frameworks such as OpenTelemetry’s emerging semantic conventions already provide foundations for standardized logging across reasoning steps and tool calls [1].
> >
> > **Second**, governance-chain logging should enable auditable reconstruction from training through deployment. This is essential when provider-imposed rules override user instructions and helps limit responsibility shifting observed in regulatory circumvention.
> >
> > **Third**, evaluation frameworks should include metrics suited to polyadic systems. Goal-consistency tests examine whether agents satisfy constraints from multiple stakeholders. Partner-steering assessments detect systematic preference for provider interests. Run-to-run variance captures non-deterministic behavior. Tools such as Microsoft’s PyRIT [2] and evaluations by NIST ARIA [3] illustrate emerging methods, but standardized third-party protocols are necessary to avoid selective testing.
> >
> > **Fourth**, mechanisms for documenting conflicts should transparently record when provider rules or incentives override user goals. Because platform-based agents centralize control with providers, structural conflicts of interest are unavoidable. One promising complementary model is the use of **agent advocates**—independent intermediaries that represent user interests in configuring, monitoring, and auditing agents [4]. They can coexist with open-source infrastructures that distribute governance power and reduce reliance on proprietary safety layers.
> >
> > These mechanisms gain legal force through documentation duties, certification conditions, and rules assigning evidentiary weight to logs and provenance records. Proposals such as California’s SB 813 illustrate how these features can be institutionalized through multi-stakeholder governance [5]. Integrating technical documentation with legal standards ensures that accountability does not depend solely on ex-post reconstruction.
> >
> > Detailed engineering specifications remain important future work beyond the scope of this paper.
>
> ---
>
> We also agree that Fig. 3 needs to be demystified. We drafted a simple decision procedure to eliminate ambiguity between five cases. Please refer to the visualization of decision tree in the below, which we would like to include in either body or Appendix based on the page limit. This helps readers map real-world scenarios to the categories with more precision. We would love to hear about reviewer’s feedback on this additional figure.
>
>  > **Figure X. Decision Tree for Classifying AI Agent Relationships**
> > Five sequential questions determine which of five possible agency configurations applies:
> > (Q1) whether the AI affects Third Parties;
> > (Q2) on whose behalf the AI acts;
> > (Q3) whether user approval is required for each action;
> > (Q4) who controls the AI's goals and constraints; and
> > (Q5) whether users can override provider constraints.
> > Most current systems fall into Case 3 (provider as agent), Case 4 (user as third party), or Case 5 (no agency), leaving users without the protections that traditional agency law would provide in Cases 1 or 2.
> >
> > ---
> >
> > **Q1. Does the AI take actions affecting Third Parties?**
> > - **No** → **Case 5** (no agency)
> >   example: ChatGPT research
> >
> > - **Yes** → Q2
> >
> > ---
> >
> > **Q2. On behalf of whom does the AI act?**
> > - **Provider / Platform** → **Case 4**
> >   example: Airline agent (User = Third Party)
> >
> > - **User** → Q3
> >
> > ---
> >
> > **Q3. Must the user approve each action?**
> > - **Yes** → **Case 5** (AI as a tool)
> >   example: Cursor without automated push
> >
> > - **No** → Q4
> >
> > ---
> >
> > **Q4. Who controls the AI’s goals/constraints?**
> > - **User** → **Case 1**
> >   example: Agent advocates (User = Principal)
> >
> > - **Provider** → Q5
> >
> > ---
> >
> > **Q5. Can the user override?**
> > - **Yes** → **Case 2**
> >   example: Investment advisor (AI as Subagent)
> >
> > - **No** → **Case 3**
> >   example: LitAI agency (Provider as sole Agent)
>
> We believe the revised manuscript responds to each of the reviewer’s concerns. The paper now complements the diagnostic analysis with concrete statutory and technical mechanisms, clarifies the evidentiary grounding of the arguments, extends the discussion beyond U.S. doctrine, and introduces operational tools such as Figure X and the accountability stack. We hope this strengthened version makes the paper’s contribution clearer.

---

> ### Author Response · Authors · 2025-11-23
> **Response 3 to Reviewer EK6J**
>
> ---
> **References**
>
> [1] Young, Ted, and Austin Parker. *Learning OpenTelemetry*. Sebastopol, CA: O’Reilly Media, 2024.
>
> [2] Munoz, Gary D. Lopez, Amanda J. Minnich, Roman Lutz, Richard Lundeen, Raja Sekhar Rao Dheekonda,
>     Nina Chikanov, Bolor-Erdene Jagdagdorj et al.
>     "PyRIT: A Framework for Security Risk Identification and Red Teaming in Generative AI Systems."
>     arXiv preprint arXiv:2410.02828 (2024).
>
> [3] Schwartz, Reva, Gabriella Waters, Razvan Amironesei, Craig Greenberg, Jon Fiscus, Patrick Hall,
>     Anya Jones et al.
>     "The Assessing Risks and Impacts of AI (ARIA) Program Evaluation Design Document." (2024).
>     https://ai-challenges.nist.gov/aria/docs/ARIA_Program_Companion_Document_Dec20.pdf
>
> [4] Kapoor, Sayash, Noam Kolt, and Seth Lazar.
>     "Position: Build Agent Advocates, Not Platform Agents."
>     In *Forty-Second International Conference on Machine Learning* (ICML) Position Paper Track, 2025.
>
> [5] Cal. S.B. 813, 2025–2026 Reg. Sess. (as amended May 23, 2025).

---

### Official Review · Reviewer_E48R · 2025-11-02

**Soundness:** 2
**Presentation:** 3
**Contribution:** 1
**Rating:** 2
**Confidence:** 4

**Summary:**

The paper discusses "key issues that hinder AI agents from attaining true legal status". It is unclear to me what the contribution is. It is not clear what the technical hinders are compared to the legal hinders from other types of hinders. At the same time the authors say "this position paper argue[s] that treating AI systems as if they were human Agents obscures fundamental structural differences..." I would agree with the second sentiment, but find it unclear how to reconcile with the formulations in the abstract. The conclusions seem to more point in the first direction, than the second.

**Strengths:**

The question of agency as well as the connection between legal and technical aspects are interesting and relevant.

The paper covers several relevant aspects to the these problems.

**Weaknesses:**

What is the actual position the paper takes? The more I read the paper, the less clear it seems to me.

It is unclear what the contributions are. I can imagine a set of technical challenges that need to be addressed or legal questions identified that must be addressed. If it is a position paper, then there needs to be a clear position (which there is) that then acts as a red thread and reaches a clear conclusion or set of arguments for the position (this is missing or unclear).

It is even unclear whether we really want agents to have legal status.I would have expected a more thorough ethical discussions of this.

See questions for more issues.

**Questions:**

Why do we want to give agents "true legal status"?

Is this mainly a technical problem? Or is it a societal problem related to acceptance (i.e. agents will have legal agency when society accepts this)?

In the conclusions it is stated that a key problem is that agents operate through "fragmented layers of control", how could this be avoided? How is this different from cars or airplanes?

When you talk about "existing legal frameworks" which ones do you refer to? Are they the same or are there some that are better/worse? Are agents legal entities in any country?

You call for developing "new institutional, technical, and legal mechanisms", are they all equally important?

---

> ### Author Response · Authors · 2025-11-23
> **Response 1 to Reviewer E48R's Feedback**
>
> We sincerely apologize for the delay. We thank the reviewer for recognizing the relevance and interest of our analysis connecting legal agency and technical aspects. We appreciate the opportunity to clarify our position and contributions.
>
> ## What is the Actual Position?
>
> The reviewer notes uncertainty about our core position. We clarify: Current AI agents should not be treated as legal Agents under existing agency law because the doctrinal fit is structurally unsound, and doing so creates false expectations about accountability while obscuring the need for new regulatory frameworks.
>
> After considerate reflections on reviewers’ feedback, we refine our position with three components:
>
> 1. **Diagnostic claim**: AI agents exhibit polyadic governance (Section 4), divided loyalty (Section 5), and accountability gaps (Section 6) that make existing agency law doctrines inadequate without substantial reinterpretation.
>
> 2. **Prescriptive claim** (newly added Section 7): Rather than stretching agency law analogies to fit AI systems, we need purpose-built statutory regimes that recognize polyadic control structures and impose ex-ante duties on multiple actors in the supply chain. Such regimes should combine legal frameworks (like the EU AI Act and revised Product Liability Directive) with technical accountability mechanisms.
>
> We will strengthen the introduction and conclusion to make this position more explicit as a red thread throughout the paper.
>
> ---
>
> > ## **Section 7. AI Agent Governance Beyond Agency Law: Legal Regimes and Technical Design**
> >
> > ### **7.1 Statutory Regimes for Polyadic Liability Allocation**
> >
> > Traditional agency law, premised on bilateral human-to-human relationships, provides little guidance when AI agents must navigate conflicting instructions from multiple stakeholders. Consider existing regulatory models that already address polyadic governance structures. The Investment Advisers Act of 1940 imposes fiduciary duties on financial advisors while recognizing that multiple parties (investment managers, broker-dealers, custodians) participate in the advisory relationship, with detailed regulations specifying each actor's boundaries of duties and potential liability.
> >
> > Similarly, California's Talent Agencies Act regulates entertainment agents by defining specific obligations for agents, personal managers, and production companies, acknowledging that talent representation involves multiple intermediaries with potentially conflicting interests.
> >
> > AI agent governance requires analogous but more granular statutory specificity corresponding to the complex nature of governance. Agency law's general principles of loyalty and care cannot adequately address situations where foundation model providers, fine-tuners, deployers, and users each impose different constraints on agent behavior.
> >
> > When an AI agent produces harm, the uncertain nature of how training decisions, safety constraints, and deployment configurations interact makes fault-based liability allocation impractical. Proving which specific actor's decision caused the harm becomes prohibitively difficult given the opacity of model behavior and the distributed nature of control.
> >
> > A more effective solution requires comprehensive legal architectures that impose ex-ante duties on multiple actors in the supply chain and distribute liability without requiring strict proof of fault. This allocation should consider three dimensions:
> > - evidentiary access to information,
> > - control over risk at different lifecycle stages,
> > - and policy considerations balancing innovation and safety.
> >
> > The EU AI Act and revised Product Liability Directive exemplify this model. The AI Act imposes differentiated obligations on providers, deployers, importers, and distributors. The revised Product Liability Directive establishes strict liability for defective products, creates rebuttable presumptions to ease plaintiff burdens of proof, and supports joint and several liability. Together, these instruments address polyadic governance by specifying actor-specific duties ex ante and enabling burden-shifting ex post.
> >
> > ---
> >

---

> ### Author Response · Authors · 2025-11-23
> **Response 2 to Reviewer E48R's feedback**
>
> > ### **7.2 Technical Governance Mechanisms**
> >
> > While statutory regimes define the legal foundation for responsibility, technical and institutional mechanisms are necessary to operationalize accountability across polyadic governance structures. We outline a minimal accountability stack that indicates feasible directions rather than prescribing full engineering specifications.
> >
> > **First**, provenance-aware documentation should record which actor shaped system behavior at each stage. Telemetry frameworks such as OpenTelemetry’s emerging semantic conventions already provide foundations for standardized logging across reasoning steps and tool calls [1].
> >
> > **Second**, governance-chain logging should enable auditable reconstruction from training through deployment. This is essential when provider-imposed rules override user instructions and helps limit responsibility shifting observed in regulatory circumvention.
> >
> > **Third**, evaluation frameworks should include metrics suited to polyadic systems. Goal-consistency tests examine whether agents satisfy constraints from multiple stakeholders. Partner-steering assessments detect systematic preference for provider interests. Run-to-run variance captures non-deterministic behavior. Tools such as Microsoft’s PyRIT [2] and evaluations by NIST ARIA [3] illustrate emerging methods, but standardized third-party protocols are necessary to avoid selective testing.
> >
> > **Fourth**, mechanisms for documenting conflicts should transparently record when provider rules or incentives override user goals. Because platform-based agents centralize control with providers, structural conflicts of interest are unavoidable. One promising complementary model is the use of **agent advocates**—independent intermediaries that represent user interests in configuring, monitoring, and auditing agents [4]. They can coexist with open-source infrastructures that distribute governance power and reduce reliance on proprietary safety layers.
> >
> > These mechanisms gain legal force through documentation duties, certification conditions, and rules assigning evidentiary weight to logs and provenance records. Proposals such as California’s SB 813 illustrate how these features can be institutionalized through multi-stakeholder governance [5]. Integrating technical documentation with legal standards ensures that accountability does not depend solely on ex-post reconstruction.
> >
> > Detailed engineering specifications remain important future work beyond the scope of this paper.
>
> ## Contributions
>
> Our contributions are:
>
> 1. **Conceptual reframing:**
>    The Agency/Loyalty/Accountability triad provides a structured framework for understanding
>    why legal agency analogies fail for AI systems.
>
> 2. **Technical–legal bridge:**
>    We connect ML failure modes (hallucination, brittleness, non-determinism) to agency law doctrines
>    (fiduciary duties, respondeat superior), showing how technical limitations create legal mismatches.
>
> 3. **Decision procedure (added):**
>    Figure X introduces an operational tool for mapping real AI systems to the five agency configurations,
>    clarifying when legal agency relationships arise and when they do not.
>
> 4. **Governance blueprint (added):**
>    Section 7 specifies how polyadic governance can be operationalized through coordinated statutory duties
>    (Section 7.1) and technical accountability mechanisms (Section 7.2).
>
> Regarding the third point, as other reviewers advocate for clarifying the reasoning behind the classification illustrated in Figure 3 (five cases of agency relationships), we have added this decision tree.
>
> > **Q1. Does the AI take actions affecting Third Parties?**
> > - **No** → **Case 5** (no agency)
> >   example: ChatGPT research
> >
> > - **Yes** → Q2
> >
> > ---
> >
> > **Q2. On behalf of whom does the AI act?**
> > - **Provider / Platform** → **Case 4**
> >   example: Airline agent (User = Third Party)
> >
> > - **User** → Q3
> >
> > ---
> >
> > **Q3. Must the user approve each action?**
> > - **Yes** → **Case 5** (AI as a tool)
> >   example: Cursor without automated push
> >
> > - **No** → Q4
> >
> > ---
> >
> > **Q4. Who controls the AI’s goals/constraints?**
> > - **User** → **Case 1**
> >   example: Agent advocates (User = Principal)
> >
> > - **Provider** → Q5
> >
> > ---
> >
> > **Q5. Can the user override?**
> > - **Yes** → **Case 2**
> >   example: Investment advisor (AI as Subagent)
> >
> > - **No** → **Case 3**
> >   example: LitAI agency (Provider as sole Agent)

---

> ### Author Response · Authors · 2025-11-23
> **Response 3 to Reviewers E48R**
>
> ## Do We Want Agents to Have Legal Status?
>
> We want to make it clear that we do **not** advocate giving AI agents legal status.
> We instead argue that expectations about “legal agency” arise because traditional legal concepts assume
> unitary control and unitary intention—assumptions that do not hold for modern AI systems.
> Our position is that structural explanations outperform metaphors of human-like agency.
>
> Section 6.2 notes that granting AI agents legal personhood would require them to bear liability,
> hold assets, and face legal consequences, none of which current systems can do.
> No jurisdiction currently recognizes AI as a legal person.
> Ghasemi (2025) [6] documents that the United States explicitly prohibits such recognition
> (e.g., Utah’s HB0249), and that Saudi Arabia’s 2017 “citizenship” for Sophia had no legal substance.
>
> Rather than creating legal personhood for AI, we argue that regulatory frameworks should allocate
> responsibility among the human and corporate actors who shape, constrain, and deploy AI systems.
>
> ## Is This Technical or Societal?
>
> This is fundamentally a **socio-technical** problem about making sense of technology within legal and institutional frameworks. The challenge is not primarily technical (e.g., making AI agents more reliable) or purely social (e.g., achieving acceptance). Rather, it concerns the mismatch between AI systems' technical characteristics (polyadic control, probabilistic behavior, distributed decision-making) and legal frameworks designed for bilateral human relationships.
>
> Society's “acceptance” of AI agents as legal actors would not solve the structural problems we identify. Even if people subjectively trust AI agents, the underlying issues remain. Our analysis shows that these issues stem from how AI systems are currently built and deployed, not from perception problems.
>
> ---
>
> ## Fragmented Control: How Is This Different from Cars or Airplanes?
>
> The reviewer’s question about how AI agents’ “fragmented layers of control” differ from the distributed manufacturing of cars or airplanes highlights an important distinction.
>
> For traditional engineered systems, **control is fragmented during design and production**, but **unified during operation**:
>
> - a driver or pilot exercises direct, real-time command,
> - component failures are treated as product defects,
> - behavior is largely deterministic and predictable.
>
> By contrast, **AI agents retain fragmented control at runtime**. Their decisions remain jointly shaped by foundation model trainers, safety-layer designers, providers, deployers, and end users. All of these actors impose constraints that continue to operate simultaneously.
>
> What counts as a “failure” is often indistinguishable from behavior intentionally shaped by one of these actors, and agent outputs are probabilistic and context-dependent rather than fixed at the moment of sale.
>
> Traditional product liability presumes that **a product’s behavior is fixed when it reaches the consumer**. AI agents undermine that assumption because their behavior is continuously updated, steered, or constrained by different actors throughout their lifecycle. This is why Section 7.1 emphasizes the need for **ex-ante duties, traceability, and rebuttable presumptions**, rather than relying solely on doctrines designed for static, manufacturer-controlled products.
>
> ---
>
> ## Which Existing Legal Frameworks?
>
> The reviewer asks which “existing legal frameworks” we reference. We clarify:
>
> - Our analysis focuses primarily on **agency law** (as articulated in the *Restatement (Third) of Agency*) as a potential legal source for AI agents.
> - Agency law doctrines—fiduciary duties, respondeat superior—were built to govern **bilateral** human Principal–Agent relationships.
> - These frameworks become inadequate without statutory supplementation.
>
> Section 6 explains these inadequacies:
>
> - **Respondeat superior** assumes bilateral employment and personal liability (Section 6.2).
> - **Fiduciary duties** presume undivided loyalty to a single principal (Section 5).
>
> Case-law adaptation alone cannot produce the ex-ante clarity required for AI governance.
>
> Section 7.1 identifies **better-suited statutory regimes** such as the EU AI Act and revised Product Liability Directive. We will revise the earlier parts of the paper to make these distinctions more explicit.
>
> ---

---

> ### Author Response · Authors · 2025-11-23
> **Response 4 to Reviewer E48R's Feedback**
>
> ## Are Institutional, Technical, and Legal Mechanisms Equally Important?
>
> These mechanisms play **complementary**, not interchangeable, roles.
>
> **Legal frameworks (Section 7.1)**
> Establish duties, liability rules, enforcement structures. Without statutory clarity, technical systems lack legal consequences.
>
> **Technical mechanisms (Section 7.2)**
> Operationalize legal duties. Provenance logging, governance-chain reconstruction, and conflict documentation make distributed control visible and auditable. Without this, legal duties cannot be enforced because plaintiffs cannot prove causation.
>
> **Institutional mechanisms**
> Provide intermediating structures—certification bodies, independent auditors, agent advocates, multistakeholder organizations—that bridge legal mandates and technical implementation.
>
> All three are necessary:
>
> - Legal duties without technical tools → unenforceable.
> - Technical tools without legal mandates → optional best practices.
> - Institutional structures without clear law or telemetry → ineffective.
>
> Section 7 presents them as integrated components. Detailed specifications remain future work.
>
> ---
>
> We believe these clarifications address the reviewer’s concerns about position clarity and demonstrate how our diagnostic analysis connects directly to concrete governance mechanisms.
>
> ---
> **References**
>
> [1] Young, Ted, and Austin Parker. *Learning OpenTelemetry*. Sebastopol, CA: O’Reilly Media, 2024.
>
> [2] Munoz, Gary D. Lopez, Amanda J. Minnich, Roman Lutz, Richard Lundeen, Raja Sekhar Rao Dheekonda,
>     Nina Chikanov, Bolor-Erdene Jagdagdorj et al.
>     "PyRIT: A Framework for Security Risk Identification and Red Teaming in Generative AI Systems."
>     arXiv preprint arXiv:2410.02828 (2024).
>
> [3] Schwartz, Reva, Gabriella Waters, Razvan Amironesei, Craig Greenberg, Jon Fiscus, Patrick Hall,
>     Anya Jones et al.
>     "The Assessing Risks and Impacts of AI (ARIA) Program Evaluation Design Document." (2024).
>     https://ai-challenges.nist.gov/aria/docs/ARIA_Program_Companion_Document_Dec20.pdf
>
> [4] Kapoor, Sayash, Noam Kolt, and Seth Lazar.
>     "Position: Build Agent Advocates, Not Platform Agents."
>     In *Forty-Second International Conference on Machine Learning* (ICML) Position Paper Track, 2025.
>
> [5] Cal. S.B. 813, 2025–2026 Reg. Sess. (as amended May 23, 2025).
>
> [6] Ghasemi, Mo. "AI and Corporate Personhood: A Comparative Analysis." SSRN 5383943 (2025).

---

### Meta-Review · Area_Chair_LeTf · 2025-12-22

**Summary:**

This position paper examines whether contemporary AI "agents" can meaningfully fit within the legal category of human agents. Reviewers agree that the topic is timely, relevant, and interdisciplinary, and that the paper presents a clear diagnosis of structural mismatches between current AI systems and legal agency doctrines. The Agency-Loyalty-Accountability framing is accessible, well written, and helpful for readers who may be unfamiliar with legal concepts. The paper’s emphasis on polyadic governance is an interesting reframing that highlights why classical notions of undivided loyalty and clear principal–agent relationships break down.

However, reviewers also find the paper’s contribution unclear and the central position insufficiently developed. Although the paper asserts that AI systems should not be treated as legal agents, it does not consistently articulate how this conclusion follows from the presented arguments, nor does it reconcile tensions between the abstract’s framing and the body’s claims. The work remains largely diagnostic and offers few concrete implications, empirical support, or paths forward. In several places the analysis relies on U.S. common-law doctrines without addressing applicability to other jurisdictions. Moreover, key claims are asserted rather than substantiated with evidence or clear normative argumentation. As a result, while the paper raises important issues, reviewers find the position under-argued and the overall contribution too diffuse for publication in its current form.

**Reviewer Concerns:**

Many of the reviewers’ concerns reflect inherent limitations of the work and are unlikely to be fully resolved in a rebuttal.

**Reviewer Scores:**

4,4,6,4

---

### Decision · Program_Chairs · 2026-01-26

Reject